# Does confidence calibration improve conformal prediction?

**Huajun Xi** *                                                         *12112806@mail.sustech.edu.cn*
*Department of Statistics and Data Science*
*Southern University of Science and Technology*

**Jianguo Huang** *                                                     *jianguo.huang@ntu.edu.sg*
*College of Computing and Data Science*
*Nanyang Technological University*

**Kangdao Liu**                                                         *kangdaoliu@gmail.com*
*Department of Computer and Information Science*
*University of Macau*

**Lei Feng**                                                            *feng_lei@sutd.edu.sg*
*Information Systems Technology and Design Pillar*
*Singapore University of Technology and Design*

**Hongxin Wei** †                                                       *weihx@sustech.edu.cn*
*Department of Statistics and Data Science*
*Southern University of Science and Technology*

**Reviewed on OpenReview:** *https://openreview.net/forum?id=6DDaTwTvdE*

## Abstract

Conformal prediction is an emerging technique for uncertainty quantification that constructs prediction sets guaranteed to contain the true label with a predefined probability. Previous works often employ temperature scaling to calibrate classifiers, assuming that confidence calibration benefits conformal prediction. However, the specific impact of confidence calibration on conformal prediction remains underexplored. In this work, we make two key discoveries about the impact of confidence calibration methods on adaptive conformal prediction. Firstly, we empirically show that current confidence calibration methods (e.g., temperature scaling) typically lead to larger prediction sets with lower confidence in adaptive conformal prediction. Secondly, by investigating the role of temperature value, we observe that high-confidence predictions produced by a low temperature lead to small prediction sets for adaptive conformal prediction. Theoretically, we prove that higher-confidence predictions with lower temperatures result in smaller prediction sets on expectation. This finding implies that the rescaling parameters in these calibration methods, when optimized with cross-entropy loss, might counteract the goal of generating small prediction sets. To address this issue, we propose **Conformal Temperature Scaling** (ConfTS), a variant of temperature scaling with a novel loss function designed to enhance the efficiency of prediction sets. This approach can be extended to optimize the parameters of other post-hoc methods of confidence calibration. Extensive experiments demonstrate that our method improves existing adaptive conformal prediction methods in both image and text classification tasks.

## 1   Introduction

Ensuring the reliability of model predictions is crucial for the safe deployment of machine learning such as autonomous driving (Bojarski et al., 2016) and medical diagnostics (Caruana et al., 2015). Numerous methods

---

*Equal Contribution

†Correspond to `weihx@sustech.edu.cn`.

have been developed to estimate uncertainty and incorporate it into predictive models, including confidence calibration (Guo et al., 2017) and Bayesian neural networks (Smith, 2013). However, these approaches do not provide formal theoretical guarantees for the reliability of model predictions. In contrast, *conformal prediction* offers a systematic approach to construct prediction sets that are theoretically guaranteed to contain the true label with a desired probability (Vovk et al., 2005; Shafer & Vovk, 2008; Balasubramanian et al., 2014; Angelopoulos & Bates, 2021). This framework thus provides trustworthiness in real-world scenarios where wrong predictions are dangerous.

In the literature, conformal prediction is frequently associated with *confidence calibration*, which expects the model to predict softmax probabilities that faithfully estimate the true correctness (Wei et al., 2022; Yuksekgonul et al., 2023; Wang, 2023; Wang et al., 2024). For example, existing conformal prediction methods usually employ temperature scaling (Guo et al., 2017), a post-hoc method that rescales the logits with a scalar temperature, for a better calibration performance (Angelopoulos et al., 2021; Lu et al., 2022; 2023; Gibbs et al., 2023). The underlying hypothesis is that well-calibrated models could yield precise probability estimates, thus enhancing the reliability of generated prediction sets. However, the rigorous impacts of current confidence calibration techniques on conformal prediction remain ambiguous in the literature, which motivates our analysis of the connection between conformal prediction and confidence calibration.

In this paper, we empirically show that existing methods of confidence calibration increase the size of prediction sets generated by adaptive conformal prediction methods (this effect does not apply to non-adaptive conformal methods such as LAC (Sadinle et al., 2019)). Moreover, we find that predictions with high confidence (rescaled with a small temperature value) tend to produce efficient prediction sets while maintaining the desired coverage guarantees. However, simply adopting an extremely small temperature value may result in meaningless prediction sets, as some tail probabilities can be truncated to zero due to the finite-precision issue. Theoretically, we prove that a smaller temperature value leads to larger non-conformity scores, resulting in more efficient prediction sets on expectation. This highlights that rescaling parameters of post-hoc calibration methods, optimized by the cross-entropy loss, might counteract the goal of generating efficient prediction sets.

To validate our theoretical findings, we propose a variant of temperature scaling, *Conformal Temperature Scaling* (ConfTS), which rectifies the optimization objective through the efficiency gap, i.e., the deviation between the threshold and the non-conformity score of the ground truth. In particular, ConfTS optimizes the temperature value by minimizing the efficiency gap. This approach can be extended to optimize the parameters of other post-hoc methods of confidence calibration, e.g., vector scaling and Platt scaling. Extensive experiments show that ConfTS can effectively enhance the efficiency of existing adaptive conformal prediction techniques, APS (Romano et al., 2020) and RAPS (Angelopoulos et al., 2021). Notably, we empirically show that post-hoc calibration methods optimized by our loss function can also improve the efficiency of prediction sets in both image and text classification (including large language models), which demonstrates the generality of our method. In addition, we provide an ablation study of loss functions to show that the proposed loss function can outperform the ConfTr loss (Stutz et al., 2022). In practice, our approach is straightforward to implement within deep learning frameworks, requiring no hyperparameter tuning and additional computational costs compared to standard temperature scaling.

We summarize our contributions as follows:

- We discover that current confidence calibration methods typically lead to larger prediction sets in adaptive conformal prediction, while high-confidence predictions (using small temperatures) can enhance the efficiency of prediction sets. We further identify a practical limitation where extremely small temperature values cause numerical precision issues.

- We provide a theoretical analysis by proving that applying smaller temperature values in temperature scaling results in more efficient prediction sets on expectation. This theoretical insight explains the relationship between confidence calibration and conformal prediction.

- We validate our theoretical findings by developing Conformal Temperature Scaling (ConfTS), a variant of temperature scaling that exploits the relationship between temperature and set efficiency.

Extensive experiments demonstrate that ConfTS enhances the efficiency of prediction sets in adaptive conformal prediction and can be extended to other post-hoc methods of confidence calibration.

## 2 Preliminary

In this work, we consider the multi-class classification task with $K$ classes. Let $\mathcal{X} \subset \mathbb{R}^d$ be the input space and $\mathcal{Y} := \{1, 2, \cdots, K\}$ be the label space. We represent a pre-trained classification model by $f : \mathcal{X} \to \mathbb{R}^K$. Let $(X, Y) \sim \mathcal{P}_{\mathcal{XY}}$ denote a random data pair sampled from a joint data distribution $\mathcal{P}_{\mathcal{XY}}$, and $\boldsymbol{f}_y(\boldsymbol{x})$ denote the $y$-th element of logits vector $\boldsymbol{f}(\boldsymbol{x})$ with an instance $\boldsymbol{x}$. Normally, the conditional probability of class $y$ is approximated by the softmax probability output $\boldsymbol{\pi}(\boldsymbol{x})$ defined as:

$$\mathbb{P}\{Y = y | X = x\} \approx \pi_y(\boldsymbol{x}; t) = \sigma(f(\boldsymbol{x}); t)_y = \frac{e^{f_y(\boldsymbol{x})/t}}{\sum_{i=1}^{K} e^{f_i(\boldsymbol{x})/t}}, \tag{1}$$

where $\sigma$ is the softmax function and $t$ denotes the temperature parameter (Guo et al., 2017). The temperature softens the output probability with $t > 1$ and sharpens the probability with $t < 1$. After training the model, the temperature can be tuned on a held-out validation set by optimization methods.

**Conformal prediction.** To provide theoretical guarantees for model predictions, conformal prediction (Vovk et al., 2005) is designated for producing prediction sets that contain ground-truth labels with a desired probability rather than predicting one-hot labels. In particular, the goal of conformal prediction is to construct a set-valued mapping $\mathcal{C} : \mathcal{X} \to 2^{\mathcal{Y}}$ that satisfies the *marginal coverage*:

$$\mathbb{P}(Y \in \mathcal{C}(X)) \geq 1 - \alpha, \tag{2}$$

where $\alpha \in (0, 1)$ denotes a user-specified error rate, and $\mathcal{C}(\boldsymbol{x}) \subset \mathcal{Y}$ is the generated prediction set. In particular, the probability is with respect to the randomness of data sample $(X, Y)$. In the following, we will use 'coverage' to represent 'marginal coverage' for convenience.

Before deployment, conformal prediction begins with a calibration step, using a held-out calibration set $\mathcal{D}_{cal} := \{(\boldsymbol{x}_i, y_i)\}_{i=1}^n$. We calculate the non-conformity score $s_i = \mathcal{S}(\boldsymbol{x}_i, y_i)$ for each example $(\boldsymbol{x}_i, y_i)$, where $s_i$ is a measure of deviation between an example and the training data, which we will specify later. Then, we determine the $1 - \alpha$ quantile of the non-conformity scores as a threshold:

$$\tau = \inf \left\{ s : \frac{|\{i : \mathcal{S}(\boldsymbol{x}_i, y_i) \leq s\}|}{n} \geq \frac{\lceil (n+1)(1-\alpha) \rceil}{n} \right\}. \tag{3}$$

For a test instance $\boldsymbol{x}_{n+1}$, we first calculate the non-conformity score for each label in $\mathcal{Y}$, and then construct the prediction set $\mathcal{C}(\boldsymbol{x}_{n+1})$ by including labels whose non-conformity score falls within $\tau$:

$$\mathcal{C}(\boldsymbol{x}_{n+1}) = \{y \in \mathcal{Y} : \mathcal{S}(\boldsymbol{x}_{n+1}, y) \leq \tau\}. \tag{4}$$

Notably, small prediction sets are often preferred. As demonstrated in previous work (Cresswell et al.), the reduction in the prediction set size has practical significance, as smaller prediction sets are more informative to enable accurate human decision making. In the following, we introduce the term '*efficiency*' to compare conformal prediction methods: a method is more *efficient* when it produces smaller prediction sets.

In this paper, we focus on *adaptive* conformal prediction methods, which are designed to improve the adaptiveness of prediction set, which requires prediction sets to communicate instance-wise uncertainty (Romano et al., 2020). However, they usually suffer from inefficiency in practice: these methods commonly produce large prediction sets (Angelopoulos et al., 2021). In particular, we take the two representative methods: APS (Romano et al., 2020) and RAPS (Angelopoulos et al., 2021).

**Adaptive Prediction Set (APS). (Romano et al., 2020)** In the APS method, the non-conformity score of a data pair $(\boldsymbol{x}, y)$ is calculated by accumulating the sorted softmax probability, defined as:

$$\mathcal{S}_{APS}(\boldsymbol{x}, y) = \pi_{(1)}(\boldsymbol{x}) + \cdots + u \cdot \pi_{o(y, \pi(\boldsymbol{x}))}(\boldsymbol{x}), \tag{5}$$

where $\pi_{(1)}(\boldsymbol{x}), \pi_{(2)}(\boldsymbol{x}), \cdots, \pi_{(K)}(\boldsymbol{x})$ are the sorted softmax probabilities in descending order, and $o(y, \pi(\boldsymbol{x}))$ denotes the order of $\pi_y(\boldsymbol{x})$, i.e., the softmax probability for the ground-truth label $y$. In addition, the term $u$ is an independent random variable that follows a uniform distribution on $[0, 1]$.

**Regularized Adaptive Prediction Set (RAPS). (Angelopoulos et al., 2021)** The non-conformity score function of RAPS encourages a small set size by adding a penalty, as formally defined below:

$$\mathcal{S}_{RAPS}(\boldsymbol{x}, y) = \pi_{(1)}(\boldsymbol{x}) + \cdots + u \cdot \pi_{o(y, \pi(\boldsymbol{x}))}(\boldsymbol{x}) + \lambda \cdot (o(y, \pi(\boldsymbol{x})) - k_{reg})^+, \tag{6}$$

where $(z)^+ = \max\{0, z\}$, $k_{reg}$ controls the number of penalized classes, and $\lambda$ is the penalty term.

Notably, both methods incorporate a uniform random variable $u$ to achieve exact $1 - \alpha$ coverage (Angelopoulos et al., 2021). Moreover, we use *coverage* and *average size* to evaluate the prediction sets. A detailed description of the metrics is provided in Appendix A.

## 3 Motivation

### 3.1 Adaptive conformal prediction with calibrated prediction

*Confidence calibration* (Guo et al., 2017) expects the model to predict softmax probabilities that faithfully estimate the true correctness: $\forall p \in [0, 1]$, $\mathbb{P}\{Y = y | \pi_y(\boldsymbol{x}) = p\} = p$. To quantify the degree of miscalibration, the Expected Calibration Error (ECE) is defined as the difference between accuracy and confidence. With $N$ samples grouped into $K$ bins $\{b_1, \cdots, b_K\}$, the ECE is calculated as:

$$\text{ECE} = \sum_{k=1}^{K} \frac{|b_k|}{N} |\text{acc}(b_k) - \text{conf}(b_k)|$$

where $\text{acc}(\cdot)$ and $\text{conf}(\cdot)$ denotes the average accuracy and confidence in bin $b_k$.

In conformal prediction, previous work claims that deep learning models are often badly miscalibrated, leading to large prediction sets that do not faithfully articulate the uncertainty of the model (Angelopoulos et al., 2021). To address the issue, researchers usually employ temperature scaling (Guo et al., 2017) to process the model outputs for better calibration performance. However, the precise impacts of current confidence calibration techniques on adaptive conformal prediction remain unexplored, which motivates our investigation into this connection.

To figure out the correlation between confidence calibration and adaptive conformal prediction, we incorporate various confidence calibration methods to adaptive conformal predictors for a ResNet50 model (He et al., 2016) on CIFAR-100 dataset (Krizhevsky et al., 2009). Specifically, we use six calibration methods, including four post-hoc methods – *vector scaling* (Guo et al., 2017), *Platt scaling* (Platt et al., 1999), *temperature scaling* (Guo et al., 2017), *Bayesian methods* (Daxberger et al., 2021), and two training methods – *label smoothing* (Szegedy et al., 2016), *mixup* (Zhang et al., 2018). More details of calibration methods and setups are presented in Appendix B and Appendix C.

**Confidence calibration methods deteriorate the efficiency of adaptive conformal prediction.** In Table 1, we present the performance of confidence calibration and conformal prediction using APS and RAPS with various calibration methods for a ResNet50 model. The results show that the influences of those calibration methods are consistent: **models calibrated by both post-hoc and training calibration techniques generate large prediction sets** with lower ECE (i.e., better calibration). For example, on the ImageNet dataset, temperature scaling enlarges the average size of prediction sets of APS from 9.06 to 12.1, while decreasing the ECE from 3.69% to 2.24%. This finding demonstrates an inverse relationship between calibration performance and prediction set efficiency. In addition, incorporating calibration methods into conformal prediction does not violate the $1 - \alpha$ marginal coverage as the assumption of data exchangeability is still satisfied: we use a hold-out validation dataset for conducting confidence calibration methods. In addition, we present the results of the LAC score (Sadinle et al., 2019) in Appendix D.1, where we observe no clear correlation between confidence calibration methods and conformal prediction.

Table 1: The performance of APS and RAPS on CIFAR-100 dataset with ResNet50 model, using various calibration methods. In particular, we apply label smoothing (LS), Mixup (Mixup), Bayesian methods (Bayesian), vector scaling (VS), Platt scaling (PS), and temperature scaling (TS). We do not employ calibration techniques in the baseline (Base). We repeat each experiment for 20 times. "↓" indicates smaller values are better. "▲" and "▼" indicate whether the performance is superior/inferior to the baseline. The results show that existing confidence calibration methods deteriorate the efficiency of APS and RAPS.

| | Method | | Base | LS | Mixup | Bayesian | TS | PS | VS |
|---|---|---|---|---|---|---|---|---|---|
| | Accuracy | | 0.77 | 0.78 | 0.78 | 0.77 | 0.77 | 0.77 | 0.77 |
| | ECE ↓ | | 8.79 | 4.39 ▲ | 2.96 ▲ | 4.30 ▲ | 3.62 ▲ | 3.81 ▲ | 4.06 ▲ |
| $\alpha = 0.1$ | APS | Coverage | 0.90 | 0.90 | 0.90 | 0.90 | 0.90 | 0.90 | 0.90 |
| | | Avg.size ↓ | 4.91 | 11.9 ▼ | 12.5 ▼ | 7.55 ▼ | 6.69 ▼ | 7.75 ▼ | 7.35 ▼ |
| | RAPS | Coverage | 0.90 | 0.90 | 0.90 | 0.90 | 0.90 | 0.90 | 0.90 |
| | | Avg.size ↓ | 2.56 | 9.50 ▼ | 10.2 ▼ | 6.46 ▼ | 3.58 ▼ | 3.72 ▼ | 3.85 ▼ |
| $\alpha = 0.05$ | APS | Coverage | 0.95 | 0.95 | 0.95 | 0.95 | 0.95 | 0.95 | 0.95 |
| | | Avg.size ↓ | 11.1 | 19.8 ▼ | 20.1 ▼ | 15.6 ▼ | 12.8 ▼ | 13.9 ▼ | 11.3 ▼ |
| | RAPS | Coverage | 0.95 | 0.95 | 0.95 | 0.95 | 0.95 | 0.95 | 0.95 |
| | | Avg.size ↓ | 6.95 | 14.5 ▼ | 15.5 ▼ | 9.34 ▼ | 10.4 ▼ | 11.0 ▼ | 8.70 ▼ |

Overall, we empirically show that current confidence calibration methods negatively impact the efficiency of prediction sets, challenging the conventional practice of employing temperature scaling in adaptive conformal prediction. While confidence calibration methods are primarily designed to address overconfidence, we conjecture that high confidence may enhance prediction sets in efficiency.

## 3.2 Adaptive conformal prediction with high-confidence prediction

In this section, we investigate how the high-confidence prediction influences the adaptive conformal prediction. In particular, we employ temperature scaling with different temperatures $t \in [0.4, 0.5, \cdots, 1.3]$ (defined in Eq. (1)) to control the confidence level. The analysis is conducted on the ImageNet dataset with various model architectures, using APS and RAPS at $\alpha = 0.1$.

**High confidence enhances the efficiency of adaptive conformal prediction.** In Figures 1a and 1b, we present the average size of prediction sets generated by APS and RAPS under various temperature values $t$. The results show that a highly-confident model, produced by a small temperature value, would decrease the average size of prediction sets. For example, using VGG16, the average size is reduced by four times – from 20 to 5, with the decrease of the temperature value from 1.3 to 0.5. In addition, we present the effect of temperature on conditional coverage in Appendix **??**. There naturally arises a question: *is it always better for efficiency to take smaller temperature values?*

In Figure 1c, we report the average size of prediction sets produced by APS on ImageNet with ResNet18, using *extremely* small temperatures (i.e. $t \in \{0.12, 0.14, \cdots, 0.2\}$). Different from the above, APS generates larger prediction sets with smaller temperatures in this range, even leading to conservative coverage. This problem stems from floating point numerical errors caused by finite precision (see Appendix E for a detailed explanation). The phenomenon indicates that it is non-trivial to find the optimal temperature value for the highest efficiency of adaptive conformal prediction.

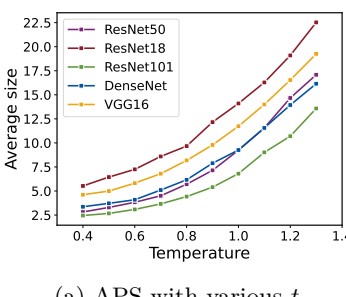 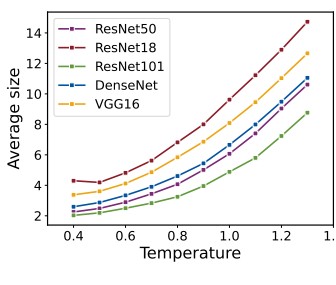 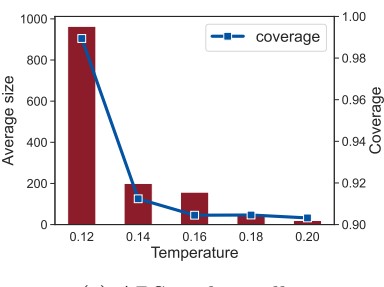

|(a) APS with various $t$|(b) RAPS with various $t$|(c) APS with small $t$|

Figure 1: (a) & (b): The performance of APS and RAPS with different temperatures on ImageNet. The results show that high-confidence predictions, with a small temperature, lead to efficient prediction sets. (Temperature softens the softmax vector with $T > 1$ and sharpens with $T < 1$.) (c): The performance of APS for ResNet18 on ImageNet with *extremely* low temperatures. In this setting, APS generates large prediction sets with conservative coverage due to finite precision.

### 3.3 Theoretical explanation

Intuitively, confident predictions are expected to yield smaller prediction sets than conservative ones. Here, we provide a theoretical justification for this by showing how the reduction of temperature decreases the average size of prediction sets in the case of non-randomized APS (simply omit the random term in Eq. (5)). We start by analyzing the relationship between the temperature $t$ and the APS score. For simplicity, assuming the logits vector $\boldsymbol{f}(\boldsymbol{x}) := [f_1(\boldsymbol{x}), f_2(\boldsymbol{x}), \dots, f_K(\boldsymbol{x})]^T$ satisfies $f_1(\boldsymbol{x}) > f_2(\boldsymbol{x}) > \cdots > f_K(\boldsymbol{x})$, then, the non-randomized APS score for class $k \in \mathcal{Y}$ is given by:

$$\mathcal{S}(\boldsymbol{x}, k, t) = \sum_{i=1}^{k} \frac{e^{f_i(\boldsymbol{x})/t}}{\sum_{j=1}^{K} e^{f_j(\boldsymbol{x})/t}}. \tag{7}$$

Then, we can derive the following proposition on the connection of the temperature and the score:

**Proposition 3.1.** *For instance $\boldsymbol{x} \in \mathcal{X}$, let $\mathcal{S}(\boldsymbol{x}, k, t)$ be the non-conformity score function of an arbitrary class $k \in \mathcal{Y}$, defined as in Eq. 7. Then, for a fixed temperature $t_0 \in \mathbb{R}^+$ and $\forall t \in (0, t_0)$, we have*

$$\mathcal{S}(\boldsymbol{x}, k, t_0) \leq \mathcal{S}(\boldsymbol{x}, k, t).$$

The proof is provided in Appendix F.1. In Proposition 3.1, we show that the APS score increases as temperature decreases, and vice versa. Then, for a fixed temperature $t_0 \in \mathbb{R}^+$, we further define $\epsilon(k, t) = \mathcal{S}(\boldsymbol{x}, k, t) - \mathcal{S}(\boldsymbol{x}, k, t_0) \geq 0$ as the difference of the APS scores. As a corollary of Proposition 3.1, we conclude that $\epsilon(k, t)$ is negatively correlated with the temperature $t$. We provide the proof for this corollary in Appendix F.2. The corollary is formally stated as follows:

**Corollary 3.2.** *For any sample $\boldsymbol{x} \in \mathcal{X}$ and a fixed temperature $t_0$, the difference $\epsilon(k, t)$ is a decreasing function with respect to $t \in (0, t_0)$.*

In the following, we further explore how the change in the APS score affects the average size of the prediction set. In the theorem, we make two continuity assumptions on the CDF of the non-conformity score (see Appendix F.3), following prior works (Lei, 2014; Sadinle et al., 2019). Given these assumptions, we can derive an upper bound for the expected size of $\mathcal{C}(\boldsymbol{x}, t)$ for any $t \in (0, t_0)$:

**Theorem 3.3.** *Under assumptions in Appendix F.3, there exists constants $c_1, \gamma \in (0, 1]$ such that*

$$\mathbb{E}_{\boldsymbol{x} \in \mathcal{X}}[|\mathcal{C}(\boldsymbol{x}, t)|] \leq K - \sum_{k \in \mathcal{Y}} c_1 [2\epsilon(k, t)]^\gamma, \quad \forall t \in (0, t_0).$$

**Interpretation.** The proof of Theorem 3.3 is presented in Appendix F.3. Through Theorem 3.3, we show that for any temperature $t$, the expected size of the prediction set $\mathcal{C}(\boldsymbol{x}, t)$ has an upper bound with respect

to the non-conformity score deviation $\epsilon$. Recalling that $\epsilon$ increases with the decrease of temperature $t$, we conclude that a lower temperature $t$ results in a larger difference $\epsilon$, thereby narrowing the prediction set $\mathcal{C}(\boldsymbol{x}, t)$. Overall, the analysis shows that tuning temperature values can potentially enhance the efficiency of adaptive conformal prediction. In practice, we may employ grid search to find the optimal $T$ for conformal prediction, but it requires defining the search range of $T$ and cannot be extended to post-hoc calibration methods with more parameters, like Platt scaling and Vector scaling. Thus, we propose an alternative solution for automatically optimizing the parameters to enhance the efficiency of conformal prediction.

### 3.4 An alternative method for improving efficiency

In the previous analysis, we empirically and theoretically demonstrate that standard temperature scaling optimized by negative log-likelihood often leads to degraded efficiency, while searching for a relatively small temperature can potentially address this issue. In this work, we propose an alternative method, Conformal Temperature Scaling (ConfTS), to automatically optimize the parameters of post-hoc calibration methods. This is a variant of temperature scaling that directly optimizes the objective function toward generating efficient prediction sets, and it can be extended to other post-hoc calibration methods.

For a test example $(\boldsymbol{x}, y)$, conformal prediction aims to construct an *efficient* prediction set $\mathcal{C}(\boldsymbol{x})$ that contains the true label $y$. Thus, the *optimal prediction set* meeting this requirement is defined as:

$$\mathcal{C}^*(\boldsymbol{x}) = \{k \in \mathcal{Y} : \mathcal{S}(\boldsymbol{x}, k) \leq \mathcal{S}(\boldsymbol{x}, y)\}.$$

Specifically, the optimal prediction set is the smallest set that allows the inclusion of the ground-truth label. Recall that the prediction set is established through the $\tau$ calculated from the calibration set (Eq. (3)), the optimal set can be attained if the threshold $\tau$ well approximates the non-conformity score of the ground-truth label $\mathcal{S}(\boldsymbol{x}, y)$. Therefore, we can measure the redundancy of the prediction set by the differences between thresholds $\tau$ and the score of true labels, defined as:

**Definition 3.4** (Efficiency Gap). *For an example $(\boldsymbol{x}, y)$, a threshold $\tau$ and a non-conformity score function $\mathcal{S}(\cdot)$, the efficiency gap of the instance $\boldsymbol{x}$ is given by:*

$$\mathcal{G}(\boldsymbol{x}, y, \tau) = \tau - \mathcal{S}(\boldsymbol{x}, y).$$

In particular, a positive efficiency gap indicates that the ground-truth label $y$ is included in the prediction set $y \in \mathcal{C}(\boldsymbol{x})$, and vice versa. To optimize for the optimal prediction set, we expect to increase the efficiency gap for samples with negative gaps and decrease it for those with positive gaps. We propose to accomplish the optimization by tuning the temperature $t$. This allows us to optimize the efficiency gap since $\mathcal{S}(\boldsymbol{x}, y)$ and $\tau$ are functions with respect to the temperature $t$ (see Eq. (7)).

**Conformal Temperature Scaling.** To this end, we propose our method – Conformal Temperature Scaling (dubbed **ConfTS**), which rectifies the objective function of temperature scaling through the efficiency gap. In particular, the loss function for ConfTS is formally given as follows:

$$\mathcal{L}_{\text{ConfTS}}(\boldsymbol{x}, y; t) = (\tau(t) - \mathcal{S}(\boldsymbol{x}, y, t))^2, \tag{8}$$

where $\tau(t)$ is the conformal threshold and $\mathcal{S}(\boldsymbol{x}, y, t)$ denotes the *non-randomized* APS score of the example $(\boldsymbol{x}, y)$ with respect to $t$ (see Eq. (7)). By minimizing the mean squared error, the ConfTS loss encourages smaller prediction sets for samples with positive efficiency gaps, and vice versa.

**The optimization of ConfTS.** To preserve the exchangeability assumption, we tune the temperature to minimize the ConfTS loss on a held-out validation set. Following previous work (Stutz et al., 2022), we split the validation set into two subsets: one to compute $\tau(t)$, and the other to calculate the ConfTS loss with the obtained $\tau(t)$. Specifically, the optimization problem can be formulated as:

$$t^* = \arg\min_{t \in \mathbb{R}^+} \frac{1}{|\mathcal{D}_{\text{loss}}|} \sum_{(\boldsymbol{x}_i, y_i) \in \mathcal{D}_{\text{loss}}} \mathcal{L}_{\text{ConfTS}}(\boldsymbol{x}_i, y_i; t), \tag{9}$$

where $\mathcal{D}_{\text{loss}}$ denotes the subset for computing ConfTS loss. Trained with the ConfTS loss, we can optimize the temperature $t$ for adaptive prediction sets with high efficiency without violating coverage. Since the

pre-defined *alpha* determines the threshold $\tau$, our ConfTS method can yield different temperature values for each $\alpha$. In addition, the $\mathcal{L}_{\text{ConfTS}}$ can be replaced by ConfTr loss (Stutz et al., 2022) or new training losses designed for different goals (e.g., improving conditional coverage). In Subsection 4.2, we show that our proposed loss is superior to ConfTr loss with improved efficiency (see Table 4).

**Extensions to other post-hoc calibration methods.** Noteworthy, our ConfTS loss is a general method and can be easily incorporated into existing post-hoc calibration methods such as Platt scaling (Platt et al., 1999) and vector scaling (Guo et al., 2017). Formally, for any rescaling function $\phi_{\boldsymbol{\theta}}$ with parameters $\boldsymbol{\theta}$, we can formulate the method as follows. First, we define the $k$-th softmax probability after rescaling as:

$$\pi_k(\boldsymbol{x}; \boldsymbol{\theta}) = \sigma(\phi_{\boldsymbol{\theta}} \cdot f(\boldsymbol{x}))_k = \frac{e^{[\phi_{\boldsymbol{\theta}} \cdot f(\boldsymbol{x})]_k}}{\sum_{i=1}^{K} e^{[\phi_{\boldsymbol{\theta}} \cdot f(\boldsymbol{x})]_i}}$$

The corresponding non-conformity score for each class $k \in \mathcal{Y}$ is given by $\mathcal{S}(\boldsymbol{x}, k; \boldsymbol{\theta}) = \sum_{i=1}^{k} \pi_k(\boldsymbol{x}; \boldsymbol{\theta})$. With the threshold $\tau(\boldsymbol{\theta})$, we rewrite the ConfTS loss by

$$\mathcal{L}_{\text{ConfTS}}(\boldsymbol{x}, y; t) = (\tau(\boldsymbol{\theta}) - \mathcal{S}(\boldsymbol{x}, y, \boldsymbol{\theta}))^2,$$

Then, the optimization objective can be formulated as:

$$\boldsymbol{\theta^*} = \arg\min_{\boldsymbol{\theta}} \frac{1}{|\mathcal{D}_{\text{loss}}|} \sum_{(\boldsymbol{x}_i, y_i) \in \mathcal{D}_{\text{loss}}} \mathcal{L}_{\text{ConfTS}}(\boldsymbol{x}_i, y_i; \boldsymbol{\theta}).$$

We present the algorithms of our proposed methods step-by-step in Appendix G. Moreover, it is worth noting that our method does not conflict with post-hoc confidence calibration, as it only changes the scaling parameters (e.g., temperature value). During inference, one may use different scaling parameters according to the objective, whether for improved calibration or smaller prediction sets.

## 4 Experiments

### 4.1 Experimental setup

**Datasets.** We evaluate ConfTS on both image and text classification tasks. For image classification, we employ CIFAR-100 (Krizhevsky et al., 2009), ImageNet (Deng et al., 2009), and ImageNet-V2 (Recht et al., 2019). For text classification, we utilize AG news (Zhang et al., 2015) and DBpedia (Auer et al., 2007) datasets. For ImageNet, we split the test dataset containing 50,000 images into 10,000 images for calibration and 40,000 for testing. For CIFAR-100 and ImageNet-V2, we split their test datasets, each containing 10,000 images, into 4,000 images for calibration and 6,000 for testing. For text datasets, we split each test dataset equally between calibration and testing. Additionally, we split the calibration set into two subsets of equal size: one subset is the validation set to optimize the temperature value with ConfTS, while the other half is the conformal set for conformal calibration.

**Models.** For ImageNet and ImageNet-V2, we employ 6 pre-trained classifiers from TorchVision (Paszke et al., 2019) – ResNet18, ResNet50, ResNet101 (He et al., 2016), DenseNet121 (Huang et al., 2017), VGG16 (Simonyan & Zisserman, 2015) and ViT-B-16 (Dosovitskiy et al., 2021). We also utilize the same model architectures for CIFAR-100 and train them from scratch. For text classification, we finetune a pre-trained BERT (Devlin, 2018) and GPT-Neo-1.3B (Black et al., 2021) on each dataset. The model architecture consists of a frozen pre-trained encoder followed by a trainable linear classifier layer. For each dataset, we employ the AdamW optimizer with a learning rate of 2e-5. The training is conducted over 3 epochs with a batch size of 32. The models are trained for 100 epochs using SGD with a momentum of 0.9, a weight decay of 0.0005, and a batch size of 128. We set the initial learning rate as 0.1 and reduce it by a factor of 5 at 60 epochs.

**Conformal prediction algorithms.** We leverage three adaptive conformal prediction methods, APS and RAPS, to generate prediction sets at error rate $\alpha \in \{0.1, 0.05\}$. In addition, we set the regularization hyperparameter for RAPS to be: $k_{reg} = 1$ and $\lambda \in \{0.001, 0.002, 0.004, 0.006, 0.01, 0.015, 0.02\}$. For the evaluation metrics, we employ *coverage* and *average size* to assess the performance of prediction sets. All experiments are repeated 20 times with different seeds, and we report average performances.

Table 2: Performance of ConfTS using APS and RAPS on ImageNet dataset. The tuned $T$ is the temperature value optimized by our loss function. We repeat each experiment 20 times. "↓" indicates that smaller values are better. **Bold** numbers are superior results. Results show that our ConfTS can improve the performance of APS and RAPS, maintaining the desired coverage rate.

| Model | Error rate | Tuned $T$ | APS | | RAPS | |
|---|---|---|---|---|---|---|
| | | | Coverage | Average size ↓ | Coverage | Average size ↓ |
| | | | | Base / ConfTS | | |
| ResNet18 | $\alpha = 0.1$ | 0.593 | 0.900 / 0.900 | 14.09 / **7.531** | 0.900 / 0.900 | 9.605 / **5.003** |
| | $\alpha = 0.05$ | 0.591 | 0.951 / 0.952 | 29.58 / **19.59** | 0.950 / 0.950 | 14.72 / **11.08** |
| ResNet50 | $\alpha = 0.1$ | 0.705 | 0.899 / 0.900 | 9.062 / **4.791** | 0.899 / 0.900 | 5.992 / **3.561** |
| | $\alpha = 0.05$ | 0.709 | 0.950 / 0.951 | 20.03 / **12.22** | 0.950 / 0.951 | 9.423 / **5.517** |
| ResNet101 | $\alpha = 0.1$ | 0.793 | 0.900 / 0.899 | 6.947 / **4.328** | 0.900 / 0.899 | 4.819 / **3.289** |
| | $\alpha = 0.05$ | 0.785 | 0.950 / 0.950 | 15.73 / **10.51** | 0.950 / 0.950 | 7.523 / **5.091** |
| DenseNet121 | $\alpha = 0.1$ | 0.659 | 0.900 / 0.899 | 9.271 / **4.746** | 0.900 / 0.900 | 6.602 / **3.667** |
| | $\alpha = 0.05$ | 0.675 | 0.950 / 0.949 | 20.37 / **11.47** | 0.949 / 0.949 | 10.39 / **6.203** |
| VGG16 | $\alpha = 0.1$ | 0.604 | 0.901 / 0.901 | 11.73 / **6.057** | 0.901 / 0.900 | 8.118 / **4.314** |
| | $\alpha = 0.05$ | 0.627 | 0.951 / 0.951 | 23.71 / **14.78** | 0.950 / 0.950 | 12.27 / **8.350** |
| ViT-B-16 | $\alpha = 0.1$ | 0.517 | 0.900 / 0.901 | 14.64 / **2.315** | 0.902 / 0.901 | 6.889 / **1.800** |
| | $\alpha = 0.05$ | 0.482 | 0.951 / 0.950 | 36.72 / **9.050** | 0.950 / 0.950 | 12.63 / **3.281** |

## 4.2 Main results

**ConfTS improves current adaptive conformal prediction methods.** In Table 2, we present the performance of APS and RAPS ($\lambda = 0.001$) with ConfTS on the ImageNet dataset. A salient observation is that ConfTS drastically improves the efficiency of adaptive conformal prediction, while maintaining the marginal coverage. For example, on the ViT model at $\alpha = 0.05$, ConfTS reduces the average size of APS by 7 times - from 36.72 to 5.759. Averaged across six models, ConfTS improves the efficiency of APS by 58.3% at $\alpha = 0.1$. We observe similar results on CIFAR-100 and ImageNet-V2 dataset in Appendix I and Appendix H. Moreover, our ConfTS remains effective for RAPS across various penalty terms on ImageNet as shown in Appendix J. Furthermore, in Appendix K, we demonstrate that ConfTS can lead to small prediction sets for *SAPS* (Huang et al., 2024), a recent technique of adaptive conformal prediction. In addition, we find that the tuned temperature values are generally smaller than 1.0 and different for various settings, which demonstrates the importance of the automatic method. Overall, empirical results show that ConfTS consistently improves the efficiency of existing adaptive conformal prediction methods.

**Our method can work with other post-hoc calibration methods.** We extend the application of ConfTS loss (Eq. (8)) to other post-hoc calibration methods. We introduce conformal Platt scaling (dubbed ConfPS) and conformal vector scaling (dubbed ConfVS), where the parameters are optimized using ConfTS loss. We employ ResNet50 and VGG16 models on the ImageNet dataset for thimage task, as well as BERT and GPT-Neo-13B on the DBpedia dataset for the text task. The error rate is $\alpha = 0.1$. Table 3 shows that both ConfPS and ConfVS can help construct efficient prediction sets. This indicates that replacing cross-entropy loss with ConfTS loss in post-hoc calibration methods consistently enhances the efficiency of adaptive conformal prediction. In addition, we provide the calibration performance of our methods in Appendix L. Overall, these results validate the effectiveness of ConfTS loss across different calibration methods.

**ConfTS maintains the adaptiveness.** Adaptiveness (Romano et al., 2020; Angelopoulos et al., 2021; Seedat et al., 2023) requires prediction sets to communicate instance-wise uncertainty: easy examples should obtain smaller sets than hard ones. In this part, we examine the impact of ConfTS on the adaptiveness of

Table 3: The average size of APS and RAPS using ConfPS and ConfVS. ConfPS and ConfVS are the variants of Platt scaling and vector scaling, optimized by our ConfTS loss. "▲" and "▼" indicate the performance is superior/inferior to the baseline. The results show that by rescaling the logits with ConfPS and ConfVS, the algorithm can construct efficient prediction sets, demonstrating the generality of our loss function.

| Dataset | Model | APS | | | | RAPS | | | |
|---------|-------|----------|--------|--------|--------|----------|--------|--------|--------|
| | | Baseline | ConfTS | ConfPS | ConfVS | Baseline | ConfTS | ConfPS | ConfVS |
| ImageNet | ResNet50 | 9.062 | 4.791 ▲ | 2.571 ▲ | 4.564 ▲ | 5.992 | 3.561 ▲ | 2.446 ▲ | 3.303 ▲ |
| | DenseNet121 | 9.271 | 4.746 ▲ | 3.169 ▲ | 5.345 ▲ | 6.602 | 3.667 ▲ | 3.224 ▲ | 3.683 ▲ |
| | VGG16 | 11.73 | 6.057 ▲ | 3.729 ▲ | 7.020 ▲ | 8.118 | 4.314 ▲ | 3.558 ▲ | 4.745 ▲ |
| | ViT-B-32 | 14.64 | 2.315 ▲ | 1.743 ▲ | 4.797 ▲ | 6.899 | 1.800 ▲ | 1.575 ▲ | 2.549 ▲ |
| AG news | BERT | 2.105 | 1.886 ▲ | 1.808 ▲ | 1.979 ▲ | 2.004 | 1.802 ▲ | 1.794 ▲ | 1.949 ▲ |
| | GPT-Neo-1.3B | 2.022 | 1.911 ▲ | 1.749 ▲ | 1.897 ▲ | 2.018 | 1.909 ▲ | 1.728 ▲ | 1.884 ▲ |
| Dbpedia | BERT | 3.557 | 2.905 ▲ | 2.96 ▲ | 3.869 ▼ | 3.458 | 2.908 ▲ | 2.837 ▲ | 3.744 ▼ |
| | GPT-Neo-1.3B | 3.171 | 2.178 ▲ | 1.826 ▲ | 1.884 ▲ | 3.137 | 2.144 ▲ | 1.768 ▲ | 2.415 ▲ |
| Average | | 13.89 | 6.697 ▲ | 4.889 ▲ | 7.839 ▲ | 9.557 | 5.526 ▲ | 4.733 ▲ | 6.068 ▲ |

Table 4: The average size of APS and RAPS with various post-hoc calibration methods optimized by our loss and ConfTr loss. **Bold** numbers are superior results between two loss functions. The results show that ConfTS loss achieves better performance than ConfTr loss in most cases.

| Model | | Baseline | ConfTS | | ConfPS | | ConfVS | |
|-------|------|----------|----------|-------------|----------|-------------|----------|-------------|
| | | | Our loss | ConfTr loss | Our loss | ConfTr loss | Our loss | ConfTr loss |
| ResNet50 | APS | 9.062 | **4.719** / | 8.864 | **2.571** / | 2.657 | 4.564 / | **4.471** |
| | RAPS | 5.992 | **3.561** / | 5.980 | **2.446** / | 2.500 | **3.303** / | 3.333 |
| VGG16 | APS | 11.73 | **6.057** / | 9.822 | **3.729** / | 4.193 | 7.020 / | **6.757** |
| | RAPS | 8.118 | **4.314** / | 6.825 | **3.558** / | 3.921 | 4.745 / | **4.742** |
| Average | | 8.726 | **4.663** / | 7.873 | **3.076** / | 3.318 | 4.908 / | **4.826** |

prediction sets and measure the instance difficulty by the order of the ground truth $o(y, \pi(\boldsymbol{x}))$. Specifically, we partition the sample by label order: 1, 2-3, 4-6, 7-10, 11-100, 101-1000, following (Angelopoulos et al., 2021). Figure 2a and Figure 2b show that prediction sets, when applied with ConfTS, satisfy the adaptiveness property. Notably, employing ConfTS can promote smaller prediction sets for all examples ranging from easy to hard. In addition, we provide a discussion on conditional coverage in Appendix **??**. Overall, the results demonstrate that APS with ConfTS succeeds in producing adaptive prediction sets: examples with lower difficulty obtain smaller prediction sets on average.

**Ablation study on the size of validation and calibration set.** In the experiment, ConfTS splits the calibration data into two subsets: validation set for tuning the temperature and conformal set for conformal calibration. In this part, we analyze the impact of this split on the performance of ConfTS by varying the validation and conformal dataset sizes from 3,000 to 8,000 samples while maintaining the other part at 5,000 samples. We use ResNet18 and ResNet50 on ImageNet, with APS at $\alpha = 0.1$. Figure 2c and 2d show that the performance of ConfTS remains consistent across different conformal dataset sizes and validation dataset sizes. Based on these results, we choose a calibration set including 10000 samples and split it into two equal subsets for the validation and conformal set. In summary, the performance of ConfTS is robust to variations in the validation dataset and conformal dataset size.

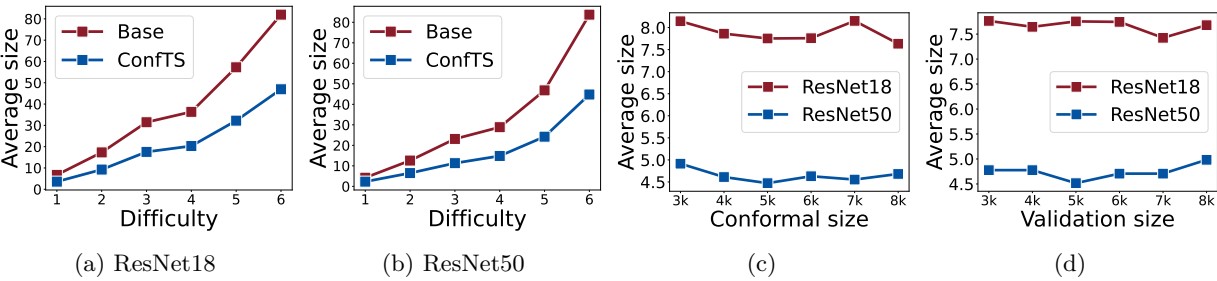

(a) ResNet18      (b) ResNet50      (c)      (d)

Figure 2: (a)&(b): Average sizes of examples with different difficulties using APS on ResNet18 and ResNet50, respectively. Results show that ConfTS can maintain adaptiveness. (c)&(d) Average sizes of APS employed with ConfTS under various sizes of (c) conformal dataset (d) validation dataset. Results show that our ConfTS is robust to variations in the validation and conformal dataset size.

Table 5: The performance of ConfTS using various non-conformity scores to compute the efficiency gap. We consider standard APS and RAPS score as well as their non-randomized variants. Each experiment is repeated 20 times. "Avg.size" and "Cov." represent the results of average size and coverage, and 'Base' presents the results without ConfTS. "↓" indicates smaller values are better. "▲" and "▼" indicate the performance is superior/inferior to the baseline. **Bold** numbers are superior results.

| Model | Score | Base | | APS_no_random | | RAPS_no_random | | APS_random | | RAPS_random | |
|---|---|---|---|---|---|---|---|---|---|---|---|
| | | Avg.size ↓ | Cov. | Avg.size ↓ | Cov. | Avg.size ↓ | Cov. | Avg.size ↓ | Cov. | Avg.size ↓ | Cov. |
| ResNet18 | APS | 14.09 | 0.900 | **7.531** ▲ | 0.900 | 7.752 ▲ | 0.900 | 13.67 ▲ | 0.900 | 13.97 ▲ | 0.900 |
| | RAPS | 9.605 | 0.900 | **5.003** ▲ | 0.900 | 5.346 ▲ | 0.900 | 11.36 ▼ | 0.900 | 11.58 ▼ | 0.900 |
| ResNet50 | APS | 9.062 | 0.900 | **4.791** ▲ | 0.900 | 5.201 ▲ | 0.900 | 12.92 ▼ | 0.900 | 16.43 ▼ | 0.900 |
| | RAPS | 5.992 | 0.900 | **3.561** ▲ | 0.900 | 3.782 ▲ | 0.900 | 9.838 ▼ | 0.900 | 11.70 ▼ | 0.900 |

**Our loss function outperforms ConfTr loss.** Previous work (Stutz et al., 2022) proposes *Conformal Training (ConfTr)*, which enhances prediction set efficiency during training through a novel ConfTr loss function. In particular, the ConfTR loss is defined as

$$\mathcal{L}_{\text{ConfTR}}(\boldsymbol{x}) = \max\left\{0, \sum_{y \in \mathcal{Y}} \sigma(\frac{1}{P}[\tau - \mathcal{S}(\boldsymbol{x}, y)]) - k\right\}$$

where $k = 1$ by default to prevent penalizing singletons and $P$ is a smoothing parameter. In this experiment, we set $P = 1.2$. We compare the performance of ConfTS, ConfPS, and ConfVS when trained with both ConfTr loss and our proposed ConfTS loss. In particular, we use the ConfTr loss to replace line 9 of the algorithms presented in Appendix G.

Using ResNet50 and VGG16 models on ImageNet, we generate prediction sets with APS and RAPS at an error rate $\alpha = 0.1$. The results in Table 4 demonstrate that while both loss functions improve prediction set efficiency, our loss typically achieves better performance than the ConfTr loss. For example, with the ResNet50 model, ConfTS loss reduces the average size of APS to 4.791, compared to 8.864 when using ConfTr loss. Overall, the proposed loss function is superior to the ConfTr loss in optimizing the calibration methods.

**Ablation study on the non-conformity score in ConfTS.** In this ablation, we compare the performance of ConfTS trained with various non-conformity scores in Eq. (8), including standard APS and RAPS, as well as their non-randomized variants. Table 5 presents the performance of prediction sets generated by standard APS and RAPS ($\lambda = 0.001$) methods with different variants of ConfTS, employing ResNet18 and ResNet50 on ImageNet. The results show that ConfTS with randomized scores fails to produce efficient prediction sets, while non-randomized scores result in small prediction sets. This is because the inclusion of the random variable $u$ leads to the wrong estimation of the efficiency gap, thereby posing challenges to the optimization process in ConfTS. Moreover, randomized APS consistently performs better than randomized RAPS, even

when using standard RAPS to generate prediction sets. Overall, our findings show that ConfTS with the non-randomized APS outperforms the other scores in enhancing the efficiency of prediction sets.

**Discussion on the conditional coverage.**
In this part, we discuss the impact of temperature on the *conditional coverage* and how ConfTS affects the conditional coverage. In particular, the conditional coverage (Vovk, 2012) requires conformal prediction methods to satisfy the marginal coverage at the instance level. The *size-stratified coverage violation (SSCV)* (Angelopoulos et al., 2021) is often employed to evaluate the conditional coverage of prediction sets:

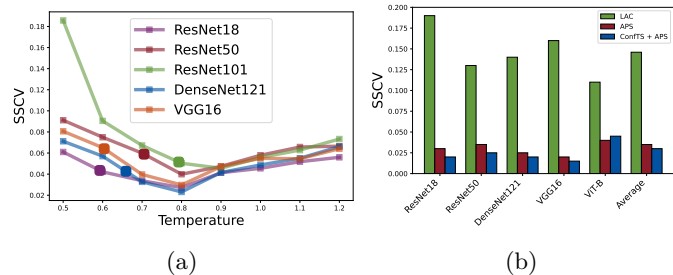

$$\text{SSCV} = \sup_j | \frac{|\{i \in S_j : y_i \in \mathcal{C}(\boldsymbol{x}_i)\}|}{|S_j|} - (1-\alpha)|,$$

where $\{S_i\}_{i=1}^{N_s}$ is a disjoint set-size strata, satisfying $\bigcup_{i=1}^{N_s} S_i = \{1, 2, \cdots, |\mathcal{Y}|\}$. A lower SSCV value indicates better conditional coverage per-

(a)          (b)

Figure 3: (a): The performance of APS on SSCV with different temperatures on ImageNet. The round marks represent the temperature value obtained from ConfTS. (b): The SSCV performance of ConfTS using APS on ImageNet dataset. A smaller SSCV is better.

formance. Following prior work (Angelopoulos et al., 2021), we set the partitioning of the set sizes as: 0-1, 2-3, 4-10, 11-100, and 101-1000. Figure 3a presents the SSCV performance of APS with varying temperatures. The results show that the optimal temperature value for the lowest SSCV of APS is generally smaller than 1.0, so vanilla temperature scaling causes worse performances on conditional coverage, as it typically encourages a large temperature. In contrast, our ConfTS usually leads to a relatively low temperature, achieving promising performance on conditional coverage.

Figure 5a presents the SSCV performance of APS with varying temperatures. The results show that the optimal temperature value for the lowest SSCV of APS is generally smaller than 1.0, so vanilla temperature scaling causes worse performances on conditional coverage, as it typically encourages a large temperature. In contrast, our ConfTS usually leads to a relatively low temperature, achieving promising performance on conditional coverage. In Figure 5b, we present a comparison of SSCV performance between LAC, APS, and APS+ConfTS on various model architectures. In this setting, the results show that our method can enhance the conditional coverage of APS, while LAC achieves the worst results on SSCV. For example, on ResNet50, ConfTS reduces the SSCV of APS from 0.033 to 0.025, and the SSCV of LAC is 0.125 – much larger than our method. Despite the empirical improvements, we emphasize that our method cannot guarantee an improved conditional coverage as it is not included in the training objective. We hope this work can inspire future work to design specific training losses to improve conditional coverage.

Based on the analysis above, we provide the following guidelines for practitioners to select the appropriate conformal technique:

- For minimal prediction set sizes with marginal coverage: use the LAC non-conformity score, which is proven to yield the smallest prediction sets, though it offers limited conditional coverage.

- For conditional coverage with compact sets: opt for APS or RAPS as the non-conformity score and apply ConfTS, ConfVS, or ConfPS to further reduce prediction set sizes (see Table 2).

## 5 Related Work

**Conformal prediction.** Conformal prediction (Papadopoulos et al., 2002; Vovk et al., 2005) is a statistical framework for uncertainty qualification. Some methods leverage post-hoc techniques to enhance prediction sets (Romano et al., 2020; Angelopoulos et al., 2021; Ghosh et al., 2023; Huang et al., 2024). For example, Adaptive Prediction Sets (APS) (Romano et al., 2020) calculates the score by accumulating the sorted softmax

values in descending order. However, the softmax probabilities typically exhibit a long-tailed distribution, and thus, those tail classes are often included in the prediction sets. To alleviate this issue, Regularized Adaptive Prediction Sets (RAPS) (Angelopoulos et al., 2021) exclude tail classes by appending a penalty to these classes, resulting in efficient prediction sets. These post-hoc methods often employ temperature scaling for better calibration performance (Angelopoulos et al., 2021; Lu et al., 2022; Gibbs et al., 2023; Lu et al., 2023). In our work, we show that existing confidence calibration methods could harm the efficiency of adaptive conformal prediction.

Some works propose training time regularizations to improve the efficiency of conformal prediction (Colombo & Vovk, 2020; Stutz et al., 2022; Einbinder et al., 2022; Bai et al.; Correia et al., 2024). For example, uncertainty-aware conformal loss function (Einbinder et al., 2022) optimizes the efficiency of prediction sets by encouraging the non-conformity scores to follow a uniform distribution. Moreover, conformal training (Stutz et al., 2022) constructs efficient prediction sets by prompting the threshold to be less than the non-conformity scores. In addition, information-based conformal training (Correia et al., 2024) incorporates side information into the construction of prediction sets. In this work, we focus on post-hoc training methods, which only require the pre-trained models for conformal prediction. ConfTS is easy to implement and requires low computational resources.

**Confidence calibration.** Confidence calibration has been studied in various contexts in recent years. Numerous methods have been developed to enhance the calibration performance of machine learning models. Some works address the miscalibration problem by post-hoc methods, including histogram binning (Zadrozny & Elkan, 2001) and Platt scaling (Platt et al., 1999). Besides, regularization methods like entropy regularization (Pereyra et al., 2017) and focal loss (Mukhoti et al., 2020) are also proposed to improve the calibration performance of deep neural networks. A concurrent work (Dabah & Tirer, 2024) also investigates the effects of temperature scaling on conformal prediction. However, they only focus on the temperature scaling and do not extend the conclusion to other post-hoc and training methods of confidence calibration. In this work, we provide a more comprehensive analysis with both post-hoc and training methods of confidence calibration. In addition to the analysis, we also provide a theoretical explanation and introduce a novel method to optimize the parameters of post-hoc calibrators automatically.

## 6 Conclusion

In this paper, we investigate the relationship between two uncertainty estimation frameworks: confidence calibration and conformal prediction. We make two discoveries about this relationship: firstly, existing confidence calibration methods would lead to larger prediction sets for adaptive conformal prediction; secondly, high-confidence prediction could enhance the efficiency of adaptive conformal prediction. We prove that applying a smaller temperature to a prediction could lead to more efficient prediction sets on expectation. Inspired by this, we propose a variant of temperature scaling, Conformal Temperature Scaling (ConfTS), which rectifies the optimization objective toward generating efficient prediction sets. Our method can be extended to other post-hoc calibrators for improving conformal predictors. Extensive experiments demonstrate that our method can enhance existing adaptive conformal prediction methods, in both image and text classification tasks. Our work challenges the conventional wisdom of utilizing confidence calibration for conformal prediction, and we hope it can inspire specially-designed methods to improve the two frameworks of uncertainty estimation.

**Limitations.** In this work, the conclusions of our analysis are mostly for adaptive conformal prediction methods, without generalizing to the LAC score. In addition, the proposed method only focuses on enhancing the efficiency of prediction sets and may not help in conditional coverage, similar to current training methods for conformal prediction. We believe it can be interesting to design loss functions specifically tailored for improving conditional coverage, in future works.

## 7 Acknowledgements

This research is supported by the Shenzhen Fundamental Research Program (Grant No. JCYJ20230807091809020). We gratefully acknowledge the support of the Center for Computational Science and Engineering at the Southern University of Science and Technology for our research.

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

## A  Conformal prediction metrics

In practice, we often use *coverage* and *average size* to evaluate prediction sets, defined as:

$$\text{Coverage} = \frac{1}{|\mathcal{D}_{test}|} \sum_{(\boldsymbol{x}_i, y_i) \in \mathcal{D}_{test}} \mathbb{1}\{y_i \in \mathcal{C}(\boldsymbol{x}_i)\}, \tag{10}$$

$$\text{Average size} = \frac{1}{|\mathcal{D}_{test}|} \sum_{(\boldsymbol{x}_i, y_i) \in \mathcal{D}_{test}} |\mathcal{C}(\boldsymbol{x}_i)|, \tag{11}$$

where $\mathbb{1}(\cdot)$ is the indicator function and $\mathcal{D}_{test}$ denotes the test dataset. The coverage rate measures the percentage of samples whose prediction set contains the true label, i.e., an empirical estimation for $\mathbb{P}\{Y \in \mathcal{C}(X)\}$. The average size measures the efficiency of prediction sets. For informative predictions (Vovk, 2012; Angelopoulos et al., 2021), the prediction sets are preferred to be efficient (i.e., small prediction sets) while satisfying the valid coverage (defined in Eq. (2)).

## B  Confidence calibration methods

Here, we briefly review three post-hoc calibration methods, whose parameters are optimized with respect to negative log-likelihood (NLL) on the calibration set, and three training calibration methods. Let $\sigma$ be the softmax function and $\boldsymbol{f} \in \mathbb{R}^K$ be an arbitrary logits vector.

**Platt Scaling (Platt et al., 1999)**  is a parametric approach for calibration. Platt Scaling learns two scalar parameters $a, b \in \mathbb{R}$ and outputs

$$\pi = \sigma(a\boldsymbol{f} + b). \tag{12}$$

**Temperature Scaling (Guo et al., 2017)**  is inspired by Platt scaling (Platt et al., 1999), using a scalar parameter $t$ for all logits vectors. Formally, for any given logits vector $\boldsymbol{f}$, the new prediction is defined by

$$\pi = \sigma(\boldsymbol{f}/t).$$

Intuitively, $t$ softens the softmax probabilities when $t > 1$ so that it alleviates over-confidence.

**Vector Scaling (Guo et al., 2017)**  is a simple extension of Platt scaling. Let $\boldsymbol{f}$ be an arbitrary logit vector, which is produced before the softmax layer. Vector scaling applies a linear transformation:

$$\pi = \sigma(M\boldsymbol{f} + b),$$

where $M \in \mathbb{R}^{K \times K}$ and $b \in \mathbb{R}^K$.

**Label Smoothing (Szegedy et al., 2016)**  softens hard labels with an introduced smoothing parameter $\alpha$ in the standard loss function (e.g., cross-entropy):

$$\mathcal{L} = -\sum_{k=1}^{K} y_i^* \log p_i, \quad y_k^* = y_k(1 - \alpha) + \alpha/K,$$

where $y_k$ is the soft label for $k$-th class. It is shown that label smoothing encourages the differences between the logits of the correct class and the logits of the incorrect class to be a constant depending on $\alpha$.

**Mixup (Zhang et al., 2018)**  is another classical work in the line of training calibration. Mixup generates synthetic samples during training by convexly combining random pairs of inputs and labels as well. To mix up two random samples $(x_i, y_i)$ and $(x_j, y_j)$, the following rules are used:

$$\bar{x} = \alpha x_i + (1 - \alpha)x_j, \quad \bar{y} = \alpha y_i + (1 - \alpha)y_j,$$

where $(\bar{x}_i, \bar{y}_i)$ is the virtual feature-target of original pairs. Previous work (Thulasidasan et al., 2019) observed that compared to the standard models, mixup-trained models are better calibrated and less prone to overconfidence in prediction on out-of-distribution and noise data.

**Bayesian Method (Daxberger et al., 2021).** Bayesian modeling provides a principled and unified approach to mitigate poor calibration and overconfidence by equipping models with robust uncertainty estimates. Specifically, Bayesian modeling handles uncertainty in neural networks by modeling the distribution over the weights. In this approach, given observed data $\mathcal{D} = \{X, y\}$, we aim to infer a posterior distribution over the model parameters $\theta$ using Bayes' theorem:

$$p(\theta|\mathcal{D}) = \frac{p(\mathcal{D}|\theta)p(\theta)}{p(\mathcal{D})}. \tag{13}$$

Here, $p(\mathcal{D}|\theta)$ represents the likelihood, $p(\theta)$ is the prior over the model parameters, and $p(\mathcal{D})$ is the evidence (marginal likelihood). However, the exact posterior $p(\theta|\mathcal{D})$ is often intractable for deep neural networks due to the high-dimensional parameter space, which makes approximate inference techniques necessary.

One common method for approximating the posterior is *Laplace approximation* (LA). The Laplace approximation assumes that the posterior is approximately Gaussian in the vicinity of the optimal parameters $\theta_{\text{MAP}}$, which simplifies inference. Mathematically, LA begins by finding the MAP estimate:

$$\theta_{\text{MAP}} = \arg\max_{\theta} \log p(\mathcal{D}|\theta) + \log p(\theta). \tag{14}$$

Then, the posterior is approximated by a Gaussian distribution:

$$p(\theta|\mathcal{D}) \approx \mathcal{N}(\theta_{\text{MAP}}, H^{-1}), \quad H = -\nabla_{\theta}^2 \log p(\theta|\mathcal{D})\Big|_{\theta=\theta_{\text{MAP}}}. \tag{15}$$

The LA provides an efficient and scalable method to capture uncertainty around the MAP estimate, making it a widely used baseline in Bayesian deep learning models.

## C  Experimental setups for motivation experiments

We conduct the experiments on CIFAR-100 (Krizhevsky et al., 2009). We split the test dataset including 10,000 images into 4,000 images for the calibration set and 6,000 for the test set. Then, we split the calibration set into two subsets of equal size: one is the validation set used for confidence calibration, while the other half is the conformal set used for conformal calibration. We train a ResNet50 model from scratch. For post-hoc methods, we train the model using standard cross-entropy loss, while for training methods, we use their corresponding specific loss functions. The training detail is presented in Section 4.1. We leverage APS and RAPS to generate prediction sets at an error rate $\alpha = 0.1$, and the hyperparameters are set to be $k_{reg} = 1$ and $\lambda = 0.001$.

## D  Experiment results of LAC

### D.1  How does confidence calibration affects LAC?

In this part, we investigate the connection between and confidence calibration methods. We employ three pre-trained classifiers: ResNet18, ResNet101 (He et al., 2016), DenseNet121 (Huang et al., 2017) on CIFAR-100, generating LAC prediction sets with $\alpha = 0.1$. In Table 6, the results show that different post-hoc methods have varying impacts on LAC prediction sets, while all of them can maintain the desired coverage rate. For example, the original average size of ResNet18 with respect to THR is 2.23, increases to 2.40 with vector scaling, 2.34 with temperature scaling, and decreases to 2.20 with Platt scaling.

Table 6: The performance of LAC prediction sets employed with different calibration methods: baseline (BS), vector scaling (VS), Platt scaling (PS), and temperature scaling (TS). ↓ indicates smaller values are better.

| Model | Metrics | BS | VS | PS | TS |
|---|---|---|---|---|---|
| ResNet18 | Avg.size ↓ | **2.23** | 2.42 | 2.26 | 2.34 |
| | Coverage | 0.90 | 0.90 | 0.90 | 0.90 |
| ResNet101 | Avg.size ↓ | 1.88 | 1.98 | **1.83** | **1.83** |
| | Coverage | 0.90 | 0.90 | 0.90 | 0.90 |
| DenseNet121 | Avg.size ↓ | 1.68 | 1.68 | 1.69 | **1.65** |
| | Coverage | 0.90 | 0.90 | 0.90 | 0.90 |

## D.2   LAC with high-confidence prediction

In Figure 4, we compare the performance of LAC prediction sets deployed with different temperatures. We observe that when used with a small temperature, models tend to generate large prediction sets, while the coverage rate stabilizes at about 0.9, maintaining the marginal coverage. Moreover, we observe that models typically construct the smallest prediction set when the temperature approximates 1. Therefore, we cannot search for an appropriate temperature that benefits LAC.

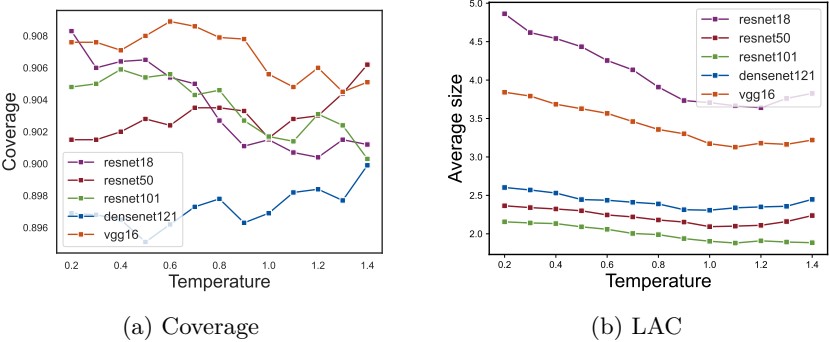

(a) Coverage                    (b) LAC

Figure 4: The performance comparison of prediction sets with different temperatures.

## E   Why numerical error occurs under an exceedingly small temperature?

In Section 3.3, we show that an exceedingly low temperature could pose challenges for prediction sets. This problem can be attributed to numerical errors. Specifically, in Proposition 3.1, we show that the softmax probability tends to concentrate in top classes with a small temperature, resulting in a long-tail distribution. Thus, the tail probabilities of some samples could be small and truncated, eventually becoming zero. For example, in Figure 5, the softmax probability is given by $\boldsymbol{\pi}(\boldsymbol{x}) = [0.999997, 2 \times 10^{-5}, 1 \times 10^{-6}, \cdots]$, and the prediction set size should be 4, following Eq. (4). However, due to numerical error, the tail probabilities, i.e., $\pi_5, \pi_6$ are truncated to be zero. This numerical error causes the conformal threshold to exceed the non-conformity scores for all classes, leading to a trivial set. Furthermore, as the temperature decreases, numerical errors occur in more data samples, resulting in increased trivial sets and consequently raising the average set size.

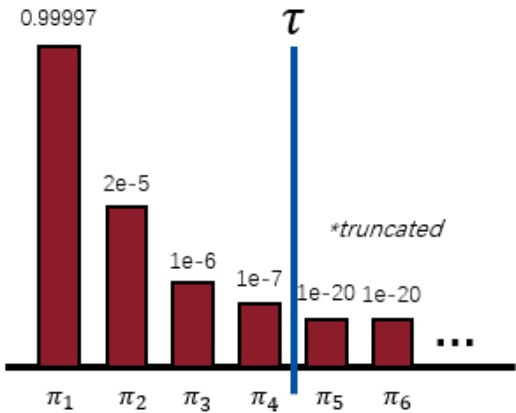

Figure 5: An example of softmax probabilities produced by a small temperature.

## F   Proofs

### F.1   Proof for Proposition 3.1

We start by showing several lemmas: the Lemma F.1, Lemma F.2 and Lemma F.3.

**Lemma F.1.** *For any given logits $(f_1, \cdots, f_K)$ with $f_1 > f_2 > \cdots > f_K$, and a constant $0 < t < 1$, we have:*

$$(a) \ \frac{e^{f_1/t}}{\sum_{i=1}^{K} e^{f_i/t}} > \frac{e^{f_1}}{\sum_{i=1}^{K} e^{f_i}},$$

$$(b) \ \frac{e^{f_K/t}}{\sum_{i=1}^{K} e^{f_i/t}} < \frac{e^{f_K}}{\sum_{i=1}^{K} e^{f_i}}.$$

*Proof.* Let $s = \frac{1}{t} - 1$. Then, we have

$$\frac{e^{f_1/t}}{\sum_{i=1}^{K} e^{f_i/t}} = \frac{e^{(1+s)f_1}}{\sum_{i=1}^{K} e^{(1+s)f_i}} = \frac{e^{f_1}}{\sum_{i=1}^{K} e^{f_i} e^{s(f_i - f_1)}} > \frac{e^{f_1}}{\sum_{i=1}^{K} e^{f_i}}.$$

$$\frac{e^{f_K/t}}{\sum_{i=1}^{K} e^{f_i/t}} = \frac{e^{(1+s)f_K}}{\sum_{i=1}^{K} e^{(1+s)f_i}} = \frac{e^{f_K}}{\sum_{i=1}^{K} e^{f_i} e^{s(f_i - f_K)}} < \frac{e^{f_1}}{\sum_{i=1}^{K} e^{f_i}}.$$

$\square$

**Lemma F.2.** *For any given logits $(f_1, \cdots, f_K)$ with $f_1 > f_2 > \cdots > f_K$, and a constant $0 < t < 1$, if there exists $j > 1$ such that*

$$\frac{e^{f_j/t}}{\sum_{i=1}^{K} e^{f_i/t}} > \frac{e^{f_j}}{\sum_{i=1}^{K} e^{f_i}},$$

*then, for all $k = 1, 2, \cdots, j$, we have*

$$\frac{e^{f_k/t}}{\sum_{i=1}^{K} e^{f_i/t}} > \frac{e^{f_k}}{\sum_{i=1}^{K} e^{f_i}}. \tag{16}$$

*Proof.* It suffices to show that

$$\frac{e^{f_{j-1}/t}}{\sum_{i=1}^{K} e^{f_i/t}} > \frac{e^{f_{j-1}}}{\sum_{i=1}^{K} e^{f_i}}, \tag{17}$$

since the rest cases where $k = 1, 2, \cdots, j - 1$ would hold by induction. The assumption gives us

$$\frac{e^{f_j/t}}{\sum_{i=1}^{K} e^{f_i/t}} > \frac{e^{f_j}}{\sum_{i=1}^{K} e^{f_i}}.$$

Let $s = \frac{1}{t} - 1$, which follows that

$$\frac{e^{f_j/t}}{\sum_{i=1}^{K} e^{f_i/t}} = \frac{e^{(1+s)f_j}}{\sum_{i=1}^{K} e^{(1+s)f_i}} = \frac{e^{f_j}}{\sum_{i=1}^{K} e^{f_i} e^{s(f_i - f_j)}} \overset{(a)}{>} \frac{e^{f_j}}{\sum_{i=1}^{K} e^{f_i}}.$$

The inequality $(a)$ indicates that

$$\sum_{i=1}^{K} e^{f_i} e^{s(f_i - f_j)} < \sum_{i=1}^{K} e^{f_i}.$$

Therefore, we can have

$$\frac{e^{f_{j-1}/t}}{\sum_{i=1}^{K} e^{f_i/t}} = \frac{e^{(1+s)f_{j-1}}}{\sum_{i=1}^{K} e^{(1+s)f_i}} = \frac{e^{f_{j-1}}}{\sum_{i=1}^{K} e^{f_i} e^{s(f_i - f_{j-1})}} > \frac{e^{f_{j-1}}}{\sum_{i=1}^{K} e^{f_i} e^{s(f_i - f_j)}} > \frac{e^{f_{j-1}}}{\sum_{i=1}^{K} e^{f_i}},$$

which proves the Eq. (17). Then, by induction, the Eq. (16) holds for all $1 \le k < j$. $\qquad \square$

**Lemma F.3.** *For any given logits* $(f_1, \cdots, f_K)$, *where* $f_1 > f_2 > \cdots > f_K$, *a constant* $0 < t < 1$, *and for all* $k = 1, 2, \cdots, K$, *we have*

$$\sum_{i=1}^{k} \frac{e^{f_i/t}}{\sum_{j=1}^{K} e^{f_j/t}} \ge \sum_{i=1}^{k} \frac{e^{f_i}}{\sum_{j=1}^{K} e^{f_j}} \tag{18}$$

*The equation holds if and only if* $k = K$.

*Proof.* The Eq. (18) holds trivially at $k = K$, since both sides are equal to 1:

$$\sum_{i=1}^{K} \frac{e^{f_i/t}}{\sum_{j=1}^{K} e^{f_j/t}} = \sum_{i=1}^{K} \frac{e^{f_i}}{\sum_{j=1}^{K} e^{f_j}} = 1, \tag{19}$$

We continue by showing the Eq. (18) at $k = K - 1$. The Lemma F.1 gives us that

$$\frac{e^{f_K/t}}{\sum_{i=1}^{K} e^{f_i/t}} < \frac{e^{f_K}}{\sum_{i=1}^{K} e^{f_i}}, \tag{20}$$

Subtracting the Eq. (20) by the Eq. (20) directly follows that

$$\sum_{i=1}^{K-1} \frac{e^{f_i/t}}{\sum_{j=1}^{K} e^{f_j/t}} > \sum_{i=1}^{K-1} \frac{e^{f_i}}{\sum_{j=1}^{K} e^{f_j}}, \tag{21}$$

which prove the Eq. (18) at $k = K - 1$. We then show that the Eq. (18) holds at $k = K - 2$, which follows that the Eq. (18) remains true for all $k = 1, 2, \cdots K - 1$ by induction. Here, we assume that

$$\sum_{i=1}^{K-2} \frac{e^{f_i/t}}{\sum_{j=1}^{K} e^{f_j/t}} < \sum_{i=1}^{K-2} \frac{e^{f_i}}{\sum_{j=1}^{K} e^{f_j}}, \tag{22}$$

and we will show that the Eq. (22) leads to a contradiction. Subtracting Eq. (22) by the Eq. (21) gives us that

$$\frac{e^{f_{K-1}/t}}{\sum_{i=1}^{K} e^{f_i/t}} > \frac{e^{f_{K-1}}}{\sum_{i=1}^{K} e^{f_i}}. \tag{23}$$

Considering the Lemma F.2, the Eq. (23) implies that

$$\frac{e^{f_k/t}}{\sum_{i=1}^{K} e^{f_i/t}} > \frac{e^{f_k}}{\sum_{i=1}^{K} e^{f_i}} \tag{24}$$

holds for all $k = 1, 2, \cdots, K - 2$. Accumulating the Eq. (24) from $k = 1$ to $K - 2$ gives us that

$$\sum_{i=1}^{K-2} \frac{e^{f_i/t}}{\sum_{j=1}^{K} e^{f_j/t}} > \sum_{i=1}^{K-2} \frac{e^{f_i}}{\sum_{j=1}^{K} e^{f_j}}.$$

This contradicts our assumption (Eq. (22)). It follows that Eq. (18) holds at $k = K - 2$. Then, by induction, the Eq. (18) remains true for all $k = 1, 2, \cdots K - 1$. Combining with the Eq. (19), we can complete our proof. $\square$

**Proposition F.4** (Restatement of Proposition 3.1). *For any sample $\boldsymbol{x} \in \mathcal{X}$, let $\mathcal{S}(\boldsymbol{x}, k, t)$ be the non-conformity score function with respect to an arbitrary class $k \in \mathcal{Y}$, defined as in Eq. 7. Then, for a fixed temperature $t_0$ and $\forall t \in (0, t_0)$, we have*

$$\mathcal{S}(\boldsymbol{x}, k, t_0) \leq \mathcal{S}(\boldsymbol{x}, k, t).$$

*Proof.* We restate the definition of non-randomized APS score in Eq. 7:

$$\mathcal{S}(\boldsymbol{x}, y, t) = \sum_{i=1}^{k} \frac{e^{f_i}}{\sum_{j=1}^{K} e^{f_j}}$$

Let $\alpha = t/t_0 \in (0, 1)$ and $\tilde{f}_i = f_i/t_0$. We rewrite the formulation of $\mathcal{S}(\boldsymbol{x}, k, t_0)$ and $\mathcal{S}(\boldsymbol{x}, k, t)$ by

$$\mathcal{S}(\boldsymbol{x}, y, t_0) = \sum_{i=1}^{k} \frac{e^{\tilde{f}_i}}{\sum_{j=1}^{K} e^{\tilde{f}_j}},$$

$$\mathcal{S}(\boldsymbol{x}, y, t) = \sum_{i=1}^{k} \frac{e^{\tilde{f}_i/\alpha}}{\sum_{j=1}^{K} e^{\tilde{f}_j/\alpha}}.$$

Since the scaling parameter $t_0$ does not change the order of $(\tilde{f}_1, \tilde{f}_2, \cdots, \tilde{f}_K)$, i.e. $\tilde{f}_1 > \tilde{f}_2 > \cdots > \tilde{f}_K$ and $\alpha \in (0, 1)$, then by the Lemma F.3, we have $\mathcal{S}(\boldsymbol{x}, y, t_0) < \mathcal{S}(\boldsymbol{x}, y, t)$. $\square$

### F.2 Proof for Corollary 3.2

**Corollary F.5** (Restatement of Corollary 3.2). *For any sample $\boldsymbol{x} \in \mathcal{X}$ and a fixed temperature $t_0$, the difference $\epsilon(k, t)$ is a decreasing function with respect to $t \in (0, t_0)$.*

*Proof.* For all $t_1, t_2$ satisfying $0 < t_1 < t_2 < t_0$, we will show that $\epsilon(k, t_1) > \epsilon(k, t_2)$. Continuing from Proposition 3.1, we have $\mathcal{S}(\boldsymbol{x}, y, t_2) < \mathcal{S}(\boldsymbol{x}, y, t_1)$. It follows that

$$\begin{aligned}
\epsilon(k, t_1) &= \mathcal{S}(\boldsymbol{x}, k, t_1) - \mathcal{S}(\boldsymbol{x}, k, t_0) \\
&> \mathcal{S}(\boldsymbol{x}, k, t_2) - \mathcal{S}(\boldsymbol{x}, k, t_0) \\
&= \epsilon(k, t_2).
\end{aligned}$$

$\square$

### F.3 Proof for Theorem 3.3

In the theorem, we make two continuity assumptions on the CDF of the non-conformity score following (Lei, 2014; Sadinle et al., 2019). We define $G_k^t(\cdot)$ as the CDF of $\mathcal{S}(\boldsymbol{x}, k, t)$, assuming that

$$
\begin{aligned}
&(1) \exists \gamma, c_1, c_2 \in (0,1] \ s.t. \ \forall k \in \mathcal{Y}, \ c_1|\varepsilon|^\gamma \leq |G_k^t(s+\varepsilon) - G_k^t(s)| \leq c_2|\varepsilon|^\gamma, \\
&(2) \exists \rho > 0 \ s.t. \ \inf_{k,s} |G_k^{t_0}(s) - G_k^t(s)| \geq \rho.
\end{aligned}
\tag{25}
$$

To prove Theorem 3.3, we start with a lemma:

**Lemma F.6.** *Give a pre-trained model, data sample $\boldsymbol{x}$, and a temperature satisfying $t^* < t_0$. Then, under assumtion (25), we have*
$$
\mathbb{P}\{k \in \mathcal{C}(\boldsymbol{x}, t_0), k \notin \mathcal{C}(\boldsymbol{x}, t^*)\} \geq c_1(2\epsilon(k, t^*))^\gamma.
$$

*Proof.* Let $\mathbb{P}^t(\cdot)$ be the probability measure corresponding to $G_y^t(\cdot)$, and $C_y^t(s) = \{x : S(\boldsymbol{x}, y, t) < s\}$. Then, we have

$$
\begin{aligned}
\mathbb{P}^{t_0}(C_y^{t_0}(\tau(t^*))) &= \mathbb{P}^{t_0}(C_y^{t^*}(\tau(t^*) + \epsilon(k, t^*)) \\
&= G_y^{t_0}(\tau(t^*) + \epsilon(k, t^*)) \\
&\overset{(a)}{\geq} G_y^{t^*}(\tau(t^*) + \epsilon(k, t^*)) + \rho.
\end{aligned}
\tag{26}
$$

where (a) comes from the assumption (2). Let $\tau^* = \tau(t^*) - \epsilon(k, t^*) - [c_2^{-1}\rho]^{1/\gamma}$. Then, replacing the $\tau(t^*)$ in Eq. (26) with $\tau^*$, we have

$$
\begin{aligned}
\mathbb{P}^{t_0}(C_y^{t_0}(\tau^*)) &\geq G_y^{t^*}(\tau(t^*) - [c_2^{-1}\rho]^{1/\gamma}) + \rho \\
&\overset{(a)}{\geq} G_y^{t^*}(\tau(t^*)) \\
&\overset{(b)}{=} \alpha \\
&\overset{(c)}{=} \mathbb{P}^{t_0}(C_y^{t_0}(\tau(t_0))).
\end{aligned}
\tag{27}
$$

where (a) is due to the assumption (1):

$$
G_y^{t^*}(\tau(t^*)) - G_y^{t^*}(\tau(t^*) - [c_2^{-1}\rho]^{1/\gamma}) \leq c_2|[c_2^{-1}\rho]^{1/\gamma}|^\gamma = \rho.
$$

(b) and (c) is because of the definition of threshold $\tau$: $C_y^{t^*}(\tau(t^*)) = C_y^{t_0}(\tau(t_0)) = \alpha$. The Eq. (28) follows that

$$
\tau(t_0) \leq \tau^* = \tau(t^*) - \epsilon(k, t^*) - [c_2^{-1}\rho]^{1/\gamma}.
\tag{28}
$$

Continuing from Eq. (28), it holds for all $y \in \mathcal{Y}$ that

$$
\begin{aligned}
\mathbb{P}\{k \in \mathcal{C}(\boldsymbol{x}, t_0), k \notin \mathcal{C}(\boldsymbol{x}, t^*)\} &\overset{(a)}{=} P\{\mathcal{S}(\boldsymbol{x}, y, t^*) < \tau(t^*), \mathcal{S}(\boldsymbol{x}, y, t_0) \geq \tau(t_0)\} \\
&\overset{(b)}{=} P\{\tau(t^*) > S(\boldsymbol{x}, y, t^*) \geq \tau(t_0) - \epsilon(k, t^*)\} \\
&\geq P\{\tau(t^*) > S(\boldsymbol{x}, y, t^*) \geq \tau(t^*) - 2\epsilon(k, t^*) - [c_2^{-1}\rho]^{1/\gamma}\} \\
&\overset{(c)}{=} G_y^{t^*}(\tau(t^*)) - G_y^{t^*}(\tau(t^*) - 2\epsilon(k, t^*) - [c_2^{-1}\rho]^{1/\gamma}) \\
&\overset{(d)}{\geq} c_1(2\epsilon(k, t^*) + [c_2^{-1}\rho]^{1/\gamma})^\gamma \\
&\geq c_1(2\epsilon(k, t^*))^\gamma.
\end{aligned}
$$

where (a) comes from the construction of prediction set: $y \in \mathcal{C}(\boldsymbol{x})$ if and only if $\mathcal{S}(\boldsymbol{x}, y) \leq \tau$. (b) is because of the definition of $\epsilon$. (c) and (d) is due to the definition of $G_y^t(\cdot)$ and assumption (1). $\qquad \square$

**Theorem F.7.** *Under the assumption equation 25, there exists constants $c_1, \gamma \in (0,1]$ such that*

$$
\mathbb{E}_{\boldsymbol{x} \in \mathcal{X}}[|\mathcal{C}(\boldsymbol{x}, t)|] \leq K - \sum_{k \in \mathcal{Y}} c_1[2\epsilon(k, t)]^\gamma, \quad \forall t \in (0, t_0).
$$

*Proof.* For all $t < t_0$, we consider the expectation size of $\mathcal{C}(\boldsymbol{x}, t)$:

$$
\begin{aligned}
\mathbb{E}_{x \in \mathcal{X}}[|\mathcal{C}(\boldsymbol{x}, t)|] &= \mathbb{E}_{x \in \mathcal{X}}\Big[\sum_{k \in \mathcal{Y}} \mathbb{1}\{k \in \mathcal{C}(\boldsymbol{x}, t)\}\Big] \\
&= \sum_{k \in \mathcal{Y}} \mathbb{E}_{x \in \mathcal{X}}[\mathbb{1}\{k \in \mathcal{C}(\boldsymbol{x}, t)\}] \\
&= \sum_{k \in \mathcal{Y}} \mathbb{P}\{k \in \mathcal{C}(\boldsymbol{x}, t)\} \\
&= \sum_{k \in \mathcal{Y}} [1 - \mathbb{P}\{k \notin \mathcal{C}(\boldsymbol{x}, t)\}].
\end{aligned}
$$

Due to the fact that

$$
\mathbb{P}\{k \in \mathcal{C}(\boldsymbol{x}, t_0), k \notin \mathcal{C}(\boldsymbol{x}, t)\} \leq \mathbb{P}\{k \notin \mathcal{C}(\boldsymbol{x}, t)\},
$$

we have

$$
\mathbb{E}_{x \in \mathcal{X}}[|\mathcal{C}(\boldsymbol{x}, t)|] \leq \sum_{k \in \mathcal{Y}} [1 - \mathbb{P}\{k \in \mathcal{C}(\boldsymbol{x}, t_0), k \notin \mathcal{C}(\boldsymbol{x}, t)\}].
$$

Continuing from Lemma F.6, we can get

$$
\mathbb{E}_{x \in \mathcal{X}}[|\mathcal{C}(\boldsymbol{x}, t)|] \leq K(1 - c_1(2\epsilon(k, t))^\gamma) = K - \sum_{k \in \mathcal{Y}} c_1(2\epsilon(k, t))^\gamma.
$$

$\square$

# G    Pseudo-algorithms of ConfTS, ConfPS and ConfVS

In this section, we present the pseudo-algorithms of the proposed methods, including ConfTS (Algorithm 1), ConfPS (Algorithm 2), and ConfVS (Algorithm 3). The essence of our method is to train a logits rescaling function with respect to the ConfTS loss. The loss function can be replaced by ConfTr or other loss functions designed for various targets.

---

**Algorithm 1** Conformal Temperature Scaling (ConfTS)

---

**Require:** Pre-trained model $f$, Validation set $D_{\text{val}} = \{(x_i, y_i)\}_{i=1}^{2n}$, Significance level $\alpha$, learning rate $\eta$, $t_{init}$, $N$

**Ensure:** Optimal temperature $t^*$

1: Split $D_{\text{val}}$ into two equal subsets $D_{\text{loss}} = \{(x_i, y_i)\}_{i=1}^{n}$ and $D_{\text{conf}} = \{(x_i, y_i)\}_{i=n}^{2n}$

2: $t \leftarrow t_{init}$

3: **for** $i = 1$ to $N$ **do**

4:    Compute calibrated probabilities: $\pi(x, y'; t) = \frac{\exp(f_{y'}(x)/t)}{\sum_{j=1}^{K} \exp(f_j(x)/t)}$   for all $y' \in \{1, \ldots, K\}$

5:    **for** each data sample $(x_i, y_i) \in D_{\text{val}}$ **do**

6:       Compute the non-randomized APS score with respect to $\pi(x, y'; t)$: $\{S_i(t)\}_{i=1}^{2n}$

7:    **end for**

8:    Compute the conformal threshold $\tau(t)$ as the $\frac{\lceil (n+1)(1-\alpha) \rceil}{n}$-quantile of scores in $D_{\text{conf}}$: $\{S_i(t)\}_{i=n}^{2n}$

9:    Compute empirical risk of ConfTS loss on $D_{\text{loss}}$:

$$\hat{R}(t) = \frac{1}{n} \sum_{i=1}^{n} \mathcal{L}_{\text{ConfTS}}(x_i, y_i; t) = \frac{1}{n} \sum_{i=1}^{n} (\tau(t) - S_i(t))^2$$

10:    $t \leftarrow t - \eta \cdot \frac{\partial}{\partial t} \hat{R}(t)$

11: **end for**

12: **return** $t$

---

**Algorithm 2** Conformal Platt Scaling (ConfPS)

---

**Require:** Pre-trained model $f$, Validation set $D_{\text{val}} = \{(x_i, y_i)\}_{i=1}^{2n}$, Significance level $\alpha$, learning rate $\eta$, $a_{init}$, $b_{init}$, $N$

**Ensure:** Optimal parameters $a^*$ and $b^*$

1: Split $D_{\text{val}}$ into two equal subsets $D_{\text{loss}} = \{(x_i, y_i)\}_{i=1}^{n}$ and $D_{\text{conf}} = \{(x_i, y_i)\}_{i=n}^{2n}$

2: $a \leftarrow a_{init}, b \leftarrow b_{init}$

3: **for** $i = 1$ to $N$ **do**

4:    Compute calibrated probabilities: $\pi(x, y'; a, b) = \frac{\exp(a \cdot f_{y'}(x) + b)}{\sum_{j=1}^{K} \exp(a \cdot f_j(x) + b)}$   for all $y' \in \{1, \ldots, K\}$

5:    **for** each data sample $(x_i, y_i) \in D_{\text{val}}$ **do**

6:       Compute the non-randomized APS score with respect to $\pi(x, y'; a, b)$: $\{S_i(a, b)\}_{i=1}^{2n}$

7:    **end for**

8:    Compute the conformal threshold $\tau(a, b)$ as the $\frac{\lceil (n+1)(1-\alpha) \rceil}{n}$-quantile of scores in $D_{\text{conf}}$: $\{S_i(a, b)\}_{i=n}^{2n}$

9:    Compute empirical risk of ConfTS loss on $D_{\text{loss}}$:

$$\hat{R}(a, b) = \frac{1}{n} \sum_{i=1}^{n} \mathcal{L}_{\text{ConfTS}}(x_i, y_i; a, b) = \frac{1}{n} \sum_{i=1}^{n} (\tau(a, b) - S_i(a, b))^2$$

10:    $a \leftarrow a - \eta \cdot \frac{\partial}{\partial a} \hat{R}(a, b), b \leftarrow b - \eta \cdot \frac{\partial}{\partial b} \hat{R}(a, b)$

11: **end for**

12: **return** $a, b$

---

---

**Algorithm 3** Conformal Vector Scaling (ConfVS)

---

**Require:** Pre-trained model $f$, Validation set $D_{\text{val}} = \{(x_i, y_i)\}_{i=1}^{2n}$, Significance level $\alpha$, learning rate $\eta$, $W_{init}$, $b_{init}$, $N$
**Ensure:** Optimal matrix $M^*$ and vector $b^*$
 1: Split $D_{\text{val}}$ into two equal subsets $D_{\text{loss}} = \{(x_i, y_i)\}_{i=1}^{n}$ and $D_{\text{conf}} = \{(x_i, y_i)\}_{i=n}^{2n}$
 2: $a \leftarrow a_{init}$, $b \leftarrow b_{init}$
 3: **for** $i = 1$ to $N$ **do**
 4:    Compute calibrated probabilities: $\pi(x, y'; M, b) = \frac{\exp(M \cdot f_{y'}(x) + b)}{\sum_{j=1}^{K} \exp(M \cdot f_j(x) + b)}$   for all $y' \in \{1, \ldots, K\}$
 5:    **for** each data sample $(x_i, y_i) \in D_{\text{val}}$ **do**
 6:       Compute the non-randomized APS score with respect to $\pi(x, y'; M, b)$: $\{S_i(M, b)\}_{i=1}^{2n}$
 7:    **end for**
 8:    Compute the conformal threshold $\tau(M, b)$ as the $\frac{\lceil (n+1)(1-\alpha) \rceil}{n}$-quantile of scores in $D_{\text{conf}}$ $\{S_i(M, b)\}_{i=n}^{2n}$
 9:    Compute empirical risk of ConfTS loss on $D_{\text{loss}}$:

$$\hat{R}(M, b) = \frac{1}{n} \sum_{i=1}^{n} \mathcal{L}_{\text{ConfTS}}(x_i, y_i; M, b) = \frac{1}{n} \sum_{i=1}^{n} (\tau(M, b) - S_i(M, b))^2$$

10:    $M \leftarrow M - \eta \cdot \frac{\partial}{\partial M} \hat{R}(M, b)$, $b \leftarrow b - \eta \cdot \frac{\partial}{\partial b} \hat{R}(M, b)$
11: **end for**
12: **return** $M, b$

---

# H   Results of ConfTS on ImageNet-V2

In this section, we show that ConfTS can effectively improve the efficiency of adaptive conformal prediction on the ImageNet-V2 dataset. In particular, we employ pre-trained ResNet50, DenseNet121, VGG16, and ViT-B-16 on ImageNet. We leverage APS and RAPS to construct prediction sets and the hyper-parameters of RAPS are set to be $k_{reg} = 1$ and $\lambda = 0.001$. In Table 7, results show that after being employed with ConfTS, APS, and RAPS tend to construct smaller prediction sets and maintain the desired coverage.

Table 7: The performance comparison of conformal prediction with baseline and ConfTS under distribution shifts. "*" denotes significant improvement (two-sample t-test at a 0.1 confidence level). "↓" indicates smaller values are better. **Bold** numbers are superior results. Results show that ConfTS can improve the efficiency of APS and RAPS on a new distribution.

| Metrics | ResNet50 | | DenseNet121 | | VGG16 | | ViT | |
|---|---|---|---|---|---|---|---|---|
| | Baseline | ConfTS | Baseline | ConfTS | Baseline | ConfTS | Baseline | ConfTS |
| Avg.size(APS) ↓ | 24.6 | **11.9*** | 50.3 | **13.3*** | 27.2 | **17.9*** | 34.2 | **10.1*** |
| Coverage(APS) | 0.90 | 0.90 | 0.90 | 0.90 | 0.90 | 0.90 | 0.90 | 0.90 |
| Avg.size(RAPS) ↓ | 13.3 | **11.3*** | 13.7 | **9.67*** | 16.3 | **13.6*** | 14.9 | **4.62*** |
| Coverage(RAPS) | 0.90 | 0.90 | 0.90 | 0.90 | 0.90 | 0.90 | 0.90 | 0.90 |

# I   Results of ConfTS on CIFAR-100

In this section, we show that ConfTS can effectively improve the efficiency of adaptive conformal prediction on the CIFAR100 dataset. In particular, we train ResNet18, ResNet50, ResNet191, ResNext50, ResNext101, DenseNet121 and VGG16 from scratch on CIFAR-100 datasets. We leverage APS and RAPS to generate prediction sets at error rates $\alpha \in \{0.1, 0.05\}$. The hyper-parameter for RAPS is set to be $k_{reg} = 1$ and $\lambda = 0.001$. In Table 8, results show that after being employed with ConfTS, APS, and RAPS tend to construct smaller prediction sets and maintain the desired coverage.

Table 8: The performance comparison of the baseline and ConfTS on CIFAR-100 dataset. We employ five models trained on CIFAR-100. "*" denotes significant improvement (two-sample t-test at a 0.1 confidence level). "↓" indicates smaller values are better. **Bold** numbers are superior results. Results show that our ConfTS can improve the performance of APS and RAPS, maintaining the desired coverage rate.

| Model | Score | $\alpha = 0.1$ | | $\alpha = 0.05$ | |
| | | Coverage | Average ↓ size | Coverage | Average size ↓ |
| | | Baseline / ConfTS | | | |
| ResNet18 | APS | 0.902 / 0.901 | 7.049 / **6.547*** | 0.949 / 0.949 | 12.58 / **11.91*** |
| | RAPS | 0.900 / 0.901 | 5.745 / **4.948*** | 0.949 / 0.949 | 8.180 / **7.689*** |
| ResNet50 | APS | 0.901 / 0.900 | 5.614 / **5.322*** | 0.951 / 0.951 | 10.27 / **10.00*** |
| | RAPS | 0.900 / 0.900 | 4.707 / **4.409*** | 0.951 / 0.950 | 7.041 / **6.811*** |
| ResNet101 | APS | 0.900 / 0.900 | 5.049 / **4.917*** | 0.949 / 0.949 | 9.520 / **9.405*** |
| | RAPS | 0.901 / 0.900 | 4.324 / **4.145*** | 0.950 / 0.950 | 6.515 / **6.450*** |
| ResNext50 | APS | 0.900 / 0.900 | 4.668 / **4.436*** | 0.950 / 0.950 | 8.911 / **8.626*** |
| | RAPS | 0.901 / 0.901 | 4.050 / **3.811*** | 0.951 / 0.951 | 6.109 / **5.854*** |
| ResNext101 | APS | 0.900 / 0.900 | 4.125 / **3.988*** | 0.950 / 0.950 | 7.801 / **7.614*** |
| | RAPS | 0.901 / 0.901 | 3.631 / **3.492*** | 0.950 / 0.950 | 5.469 / **5.253*** |
| DenseNet121 | APS | 0.899 / 0.899 | 4.401 / **3.901*** | 0.949 / 0.949 | 8.364 / **7.592*** |
| | RAPS | 0.898 / 0.898 | 3.961 / **3.434*** | 0.950 / 0.949 | 6.336 / **5.222*** |
| VGG16 | APS | 0.900 / 0.900 | 7.681 / **6.658*** | 0.949 / 0.950 | 12.36 / **11.70*** |
| | RAPS | 0.899 / 0.900 | 6.826 / **5.304*** | 0.949 / 0.949 | **10.32*** / 11.70 |

## J   Results of ConfTS on RAPS with various penalty terms

Recall that the RAPS method modifies APS by including a penalty term $\lambda$ (see Eq. (6)). In this section, we investigate the performance of ConfTS on RAPS with various penalty terms. In particular, we employ the same model architectures with the main experiment on ImageNet (see Section 4.1) and generate prediction sets with RAPS ($k_{reg} = 1$) at an error rate $\alpha = 0.1$, varying the penalty $\lambda \in \{0.002, 0.004, 0.006, 0.01, 0.015, 0.02\}$ and setting $k_{reg}$ to 1. Table 9 and 10 show that our ConfTS can enhance the efficiency of RAPS across various penalty values.

Table 9: The performance of ConfTS on RAPS with various penalty terms $\lambda \in \{0.002, 0.004, 0.006\}$ at ImageNet. "*" denotes significant improvement (two-sample t-test at a 0.1 confidence level). "↓" indicates smaller values are better. **Bold** numbers are superior results. Results show that our ConfTS can enhance the efficiency of RAPS across various penalty values.

| Model | $\lambda = 0.002$ | | $\lambda = 0.004$ | | $\lambda = 0.006$ | |
| | Coverage | Average size ↓ | Coverage | Average size ↓ | Coverage | Average size ↓ |
| | Baseline / ConfTS | | | | | |
| ResNet18 | 0.901 / 0.900 | 8.273 / **4.517*** | 0.901 / 0.901 | 6.861 / **4.319*** | 0.901 / 0.901 | 6.109 / **4.282*** |
| ResNet50 | 0.899 / 0.900 | 5.097 / **3.231*** | 0.899 / 0.900 | 4.272 / **2.892*** | 0.899 / 0.900 | 3.858 / **2.703*** |
| ResNet101 | 0.900 / 0.900 | 4.190 / **2.987*** | 0.901 / 0.899 | 3.599 / **2.686*** | 0.900 / 0.900 | 3.267 / **2.516*** |
| DenseNet121 | 0.901 / 0.901 | 5.780 / **3.340*** | 0.900 / 0.900 | 4.888 / **3.014*** | 0.900 / 0.900 | 4.408 / **2.836*** |
| VGG16 | 0.901 / 0.900 | 7.030 / **3.902*** | 0.901 / 0.900 | 5.864 / **3.514*** | 0.901 / 0.900 | 5.241 / **3.344*** |
| ViT-B-16 | 0.901 / 0.900 | 5.308 / **1.731*** | 0.901 / 0.901 | 4.023 / **1.655*** | 0.901 / 0.901 | 3.453 / **1.611*** |

Table 10: The performance of ConfTS on RAPS with various penalty terms $\lambda \in \{0.01, 0.015, 0.02\}$ at ImageNet. "*" denotes significant improvement (two-sample t-test at a 0.1 confidence level). "↓" indicates smaller values are better. **Bold** numbers are superior results. Results show that our ConfTS can enhance the efficiency of RAPS across various penalty values.

| Model | $\lambda = 0.01$ | | $\lambda = 0.015$ | | $\lambda = 0.02$ | |
|---|---|---|---|---|---|---|
| | Coverage | Average size ↓ | Coverage | Average size ↓ | Coverage | Average size ↓ |
| | Baseline / ConfTS | | | | | |
| ResNet18 | 0.901 / 0.901 | 5.281 / **4.449**\* | 0.901 / 0.901 | 4.712 / **4.683**\* | 0.900 / 0.900 | **4.452**\* / 4.917 |
| ResNet50 | 0.899 / 0.900 | 3.380 / **2.505**\* | 0.900 / 0.901 | 3.048 / **2.373**\* | 0.901 / 0.901 | 2.860 / **2.321**\* |
| ResNet101 | 0.900 / 0.900 | 2.902 / **2.317**\* | 0.900 / 0.899 | 2.643 / **2.168**\* | 0.900 / 0.900 | 2.484 / **2.096**\* |
| DenseNet121 | 0.900 / 0.900 | 3.843 / **2.657**\* | 0.900 / 0.900 | 3.452 / **2.587**\* | 0.901 / 0.899 | 3.213 / **2.750**\* |
| VGG16 | 0.900 / 0.900 | 4.537 / **3.371**\* | 0.900 / 0.900 | 4.060 / **3.423**\* | 0.899 / 0.899 | 3.744 / **3.530**\* |
| ViT-B-16 | 0.901 / 0.900 | 2.872 / **1.564**\* | 0.901 / 0.900 | 2.508 / **1.543**\* | 0.900 / 0.900 | 2.285 / **1.535**\* |

## K  Results of ConfTS on SAPS

Recall that APS calculates the non-conformity score by accumulating the sorted softmax values in descending order. However, the softmax probabilities typically exhibit a long-tailed distribution, allowing for easy inclusion of those tail classes in the prediction sets. To alleviate this issue, *Sorted Adaptive Prediction Sets (SAPS)* (Huang et al., 2024) discards all the probability values except for the maximum softmax probability when computing the non-conformity score. Formally, the non-conformity score of SAPS for a data pair $(\boldsymbol{x}, y)$ can be calculated as

$$S_{saps}(\boldsymbol{x}, y, u; \hat{\pi}) := \begin{cases} u \cdot \hat{\pi}_{max}(\boldsymbol{x}), & \text{if } o(y, \hat{\pi}(\boldsymbol{x})) = 1, \\ \hat{\pi}_{max}(\boldsymbol{x}) + (o(y, \hat{\pi}(\boldsymbol{x})) - 2 + u) \cdot \lambda, & \text{else}, \end{cases}$$

where $\lambda$ is a hyperparameter representing the weight of ranking information, $\hat{\pi}_{max}(\boldsymbol{x})$ denotes the maximum softmax probability and $u$ is a uniform random variable.

In this section, we investigate the performance of ConfTS on SAPS with various weight terms. In particular, we employ the same model architectures with the main experiment on ImageNet (see Section 4.1) and generate prediction sets with SAPS at an error rate $\alpha = 0.1$, varying the weight $\lambda \in \{0.01, 0.02, 0.03, 0.05, 0.1, 0.12\}$. Table 11 and Table 12 show that our ConfTS can enhance the efficiency of SAPS across various weights.

Table 11: The Performance of ConfTS on SAPS with various penalty terms $\lambda \in [0.005, 0.01, 0.02]$. "*" denotes significant improvement (two-sample t-test at a 0.1 confidence level). "↓" indicates smaller values are better. **Bold** numbers are superior results. Results show that our ConfTS can enhance the efficiency of SAPS across various penalty values.

| Model | $\lambda = 0.005$ | | $\lambda = 0.01$ | | $\lambda = 0.02$ | |
|---|---|---|---|---|---|---|
| | Coverage | Average size ↓ | Coverage | Average size ↓ | Coverage | Average size ↓ |
| | Baseline / ConfTS | | | | | |
| ResNet18 | 0.901 / 0.900 | 37.03 / **27.38**\* | 0.901 / 0.902 | 19.91 / **14.81**\* | 0.900 / 0.901 | 11.21 / **8.469**\* |
| ResNet50 | 0.899 / 0.899 | 27.13 / **21.37**\* | 0.899 / 0.899 | 14.45 / **11.48**\* | 0.899 / 0.899 | 8.016 / **6.510**\* |
| ResNet101 | 0.901 / 0.901 | 24.89 / **20.78**\* | 0.901 / 0.901 | 13.21 / **11.16**\* | 0.901 / 0.901 | 7.350 / **6.287**\* |
| DenseNet121 | 0.900 / 0.901 | 30.54 / **22.67**\* | 0.900 / 0.901 | 16.28 / **12.30**\* | 0.901 / 0.901 | 9.085 / **6.968**\* |
| VGG16 | 0.900 / 0.900 | 34.88 / **25.57**\* | 0.900 / 0.900 | 18.56 / **13.71**\* | 0.901 / 0.900 | 10.34 / **7.788**\* |
| ViT-B-16 | 0.901 / 0.900 | 18.90 / **11.51**\* | 0.901 / 0.900 | 10.11 / **6.379**\* | 0.900 / 0.900 | 5.669 / **3.784**\* |
| Average | 0.900 / 0.900 | 28.89 / **21.54**\* | 0.900 / 0.900 | 15.42 / **11.63**\* | 0.900 / 0.900 | 8.611 / **6.634**\* |

Table 12: The performance of ConfTS on SAPS with various penalty terms $\lambda \in \{0.03, 0.05, 0.1\}$. "*" denotes significant improvement (two-sample t-test at a 0.1 confidence level). "↓" indicates smaller values are better. **Bold** numbers are superior results. Results show that our ConfTS can enhance the efficiency of SAPS across various penalty values.

| Model | $\lambda = 0.03$ | | $\lambda = 0.05$ | | $\lambda = 0.1$ | |
|---|---|---|---|---|---|---|
| | Coverage | Average size ↓ | Coverage | Average size ↓ | Coverage | Average size ↓ |
| | | | Baseline / ConfTS | | | |
| ResNet18 | 0.900 / 0.900 | 8.206 / **6.269**\* | 0.900 / 0.900 | 5.747 / **4.716**\* | 0.901 / 0.901 | **4.143**\* / 4.581 |
| ResNet50 | 0.899 / 0.899 | 5.853 / **4.838**\* | 0.899 / 0.900 | 4.122 / **3.464**\* | 0.899 / 0.900 | 2.753 / **2.460**\* |
| ResNet101 | 0.901 / 0.901 | 5.364 / **4.640**\* | 0.901 / 0.901 | 3.756 / **3.293**\* | 0.899 / 0.900 | 2.511 / **2.286**\* |
| DenseNet121 | 0.900 / 0.900 | 6.600 / **5.151**\* | 0.900 / 0.900 | 4.601 / **3.672**\* | 0.900 / 0.900 | 3.063 / **2.811**\* |
| VGG16 | 0.900 / 0.900 | 7.504 / **5.785**\* | 0.900 / 0.900 | 5.225 / **4.173**\* | 0.900 / 0.900 | **3.483**\* / 3.551 |
| ViT-B-16 | 0.900 / 0.900 | 4.197 / **2.905**\* | 0.900 / 0.900 | 2.995 / **2.212**\* | 0.901 / 0.900 | 2.114 / **1.768**\* |
| Average | 0.900 / 0.900 | 6.287 / **4.931**\* | 0.900 / 0.900 | 4.407 / **3.588**\* | 0.900 / 0.900 | 3.011 / **2.909**\* |

## L   Calibration performance of ConfTS, ConfPS, and ConfVS

In this part, we demonstrate the tradeoff between the ECE and the prediction set size by comparing standard scaling methods — temperature scaling (TS), Platt scaling (PS), and vector scaling (VS) — to their conformal variants - ConfTS, ConfPS, and ConfVS. We conduct the experiment on ImageNet dataset, with ResNet50 and DenseNet121. We generate prediction sets with APS score and error rate $\alpha = 0.1$. The results in Table 13 show that TS, PS, and VS generate large prediction sets with lower ECE (i.e., better calibration) that are aligned with our finding in Section 3.1. In contrast, ConfTS, ConfPS, and ConfVS essentially reduce the prediction set size with higher ECE. The results are intuitive as our methods are designed for conformal prediction, instead of confidence calibration. Importantly, our method does not conflict with confidence calibration, as it only replaces the temperature value. During inference, one may use different temperature values according to the objective, whether for improved calibration performance or efficient prediction sets.

Table 13: The ECE results of ConfTS, ConfPS and ConfVS on ImageNet dataset.

| Model | ResNet50 | | | DenseNet121 | | |
|---|---|---|---|---|---|---|
| Method | Tuned T | ECE | Avg.size | Tuned T | ECE | Avg.size |
| Base | 1.000 | 4.36 | 9.062 | 1.000 | 3.66 | 9.271 |
| TS | 1.140 | 3.02 | 12.29 | 1.081 | 2.93 | 12.06 |
| ConfTS | 0.705 | 47.78 | 4.791 | 0.659 | 45.63 | 4.746 |
| PS | - | 2.55 | 12.49 | - | 2.34 | 10.72 |
| ConfPS | - | 19.23 | 2.57 | - | 21.6 | 3.169 |
| VS | - | 2.44 | 11.29 | - | 2.69 | 11.33 |
| ConfVS | - | 12.05 | 4.56 | - | 12.9 | 5.345 |

