# OpenReview forum: "Does confidence calibration improve conformal prediction?"
_TMLR — Accepted by TMLR_

### Review · Reviewer_FtzM · 2025-04-05

**Summary Of Contributions:**

This paper investigates the affect of temperature scaling (TS) on the average set size of conformal prediction sets generated using the APS and RAPS score functions. The authors find that TS with lower temperature values generally results in smaller sets on average (the only exception is at very small temperature values at which point numerical precision becomes an issue). They then propose an alternative to post-processing softmax vectors before applying APS: rather than apply TS, one can apply “Conformal Temperature Scaling” which optimizes the TS parameter to minimize the “efficiency gap,” leading to smaller set sizes. They present empirical results on several image and NLP datasets.

**Audience:**

Yes

**Claims And Evidence:**

No

**Requested Changes:**

Critical:
1. There should be some attempt to compute X-conditional coverage and consideration of X-conditional coverage should be included from the beginning. Currently, the paper presents reductions in set size as a unilateral gain without consideration of the tradeoff with X-conditional coverage. The problem that is studied is an interesting one, but the paper in its current state does not capture the full story.

2. It is not obvious to me that Conformal Temperature Scaling (Eq 8) actually yields calibration. Do you have a table that reports the ECE after applying ConfTS? If ECE is not improved by ConfTS relative to no-calibration, then it should be emphasize that ConfTS is not being proposed as a general purpose alternative to TS but rather as a pre-processing step only applicable before applying APS/RAPS.

Small comments:
* To aid in interpretation of Figure 1, perhaps there could be a brief remark about what is the range of temperature values that sharpens vs. flattens the softmax vector, relative to no calibration.

**Strengths And Weaknesses:**

Strengths:
* The paper is well-written and easy to follow
* The figures and tables are presented in a way that is easy to interpret

Weaknesses:
* The paper currently ignores a key motivation for using adaptive score functions like APS and RAPS—namely, their ability to achieve approximate X-conditional coverage. There is no discussion or computation of X-conditional coverage, even in a rough way (e.g., defining groups in feature space and computing group-conditional coverage). If you are in a setting where you do not care about X-conditional coverage and only care about set size and marginal coverage, then you could just use 1-softmax (what the paper calls the LAC score) as your conformal score function, as this score tends to give smaller sets than APS or RAPS.

---

> ### Author Response · Authors · 2025-04-26
>
> We thank the reviewer for the feedback.
>
>
> > 1. Discussion of X-conditional coverage.
>
>
> Thank you for raising the concern. In Appendix M, we provide an additional discussion on conditional coverage. In particular, we empirically show that our method can decrease the SSCV of APS with error rate $\alpha=0.05$ using various models. Despite the empirical improvements, we emphasize that our method cannot guarantee an improved conditional coverage as it is not included in the training objective. We hope this work can inspire future work to design specific training losses to improve conditional coverage.
>
> > 2. The paper does not provide results on expected calibration error (ECE) after applying Conformal Temperature Scaling (ConfTS).
>
> Thank you for raising the concern. As suggested by two reviewers (FtzM and UXE6), we present the ECE results of ConfTS, ConfPS and ConfVS in Appendix L. The results in Table 13 show that the parameters optimized by our methods achieve much smaller prediction sets than the baseline. It is also intuitive to see that the original version of TS, PS, and VS achieves lower ECE than the baseline and our methods, as our methods are not designed for confidence calibration. Notably, these two objectives do not conflict in practice as we can use different values of parameters (e.g., temperature) according to our goals in inference. To avoid any misunderstanding, we also include a clarification at the end of Section 3.
>
> > 3. Figure 1 should include a remark about temperature.
>
> Thank you for the suggestion. In the revised version, we update the caption of Figure 1 with an additional description: temperature scaling softens the softmax vector with $T>1$ and sharpens with $T<1$.
>
> ### References
> [1] Angelopoulos A, et al. Uncertainty sets for image classifiers using conformal prediction. ICLR, 2021.

---

> > ### Comment · Reviewer_FtzM · 2025-04-27
> > **Practical takeaways**
> >
> > Thank you to the authors for adding these new results. I find Figure 5(b) in Appendix M to be compelling. To me, the full story is *almost* apparent. In this paper, the authors have essentially proposed two new conformal score functions: ConfTS + APS and ConfTS + RAPS. So far, comparisons have been made to raw APS and raw RAPS, but it would also be useful to make direct comparisons to the other most common score function, namely LAC (1-softmax). It might be nice to include LAC in Figure 5(b) as well (I would expect it to be the worst, but still useful to confirm). I would encourage the authors to view the paper through the lens of a practitioner who wants to decide which conformal score function is appropriate for their setting and perhaps add a paragraph about this. e.g.,
> > * If I only care about marginal coverage and having small sets, what should I use? (My guess: LAC)
> > * If I care about adaptivity/conditional coverage and having small sets, what should I use? (My guess: ConfTS + APS or ConfTS + RAPS)

---

> > > ### Author Response · Authors · 2025-04-28
> > >
> > > Thank you for the great suggestion. As you suggested, we add the SSCV results of LAC in Figure 5(b). Indeed, LAC performs much worse than APS and ConfTS+APS on SSCV, which indicates the advantage of our method compared to LAC on conditional coverage. In addition, we provide a practical guideline for practitioners to choose a suitable conformal technique, highlighting ConfTS as an effective method for reducing the prediction set sizes in adaptive conformal prediction approaches. Thank you again for your valuable feedback, which has significantly enhanced the quality of this work.

---

### Review · Reviewer_gbFS · 2025-04-14

**Summary Of Contributions:**

## Summary of Contributions

The paper investigates the commonly held practice of applying **confidence calibration** methods (e.g., temperature scaling) prior to performing **adaptive conformal prediction (CP)**. It challenges this convention by demonstrating, both empirically and theoretically, that such calibration methods can **adversely impact the efficiency** (i.e., tightness) of prediction sets in CP, even while maintaining conditional (?) coverage.

To address this, the authors propose a novel loss function — **Conformal Temperature Scaling (ConfTS)** — which directly optimizes for the tightness of the prediction set. This loss function replaces the typical negative log-likelihood used to select temperature parameters in post-hoc calibration methods.

Importantly, this approach is **modular** (though there is missing emphasis on this) and can be integrated into **temperature scaling, Platt scaling, and vector scaling**. When used as a precursor to CP, these modified calibrators result in significantly **tighter prediction sets** without sacrificing coverage.

Extensive experiments across multiple datasets and model architectures in both image and text classification settings support the tightness of their method. The paper also presents theoretical insights and ablation studies.

**Audience:**

Yes

**Claims And Evidence:**

Yes

**Requested Changes:**

I regard responses to these as crucial, but I am open to understanding why some of my proposed changes should not be actioned.

### Abstract
- It needs to be made more clear that current calibration methods result in higher temperatures, which lead to larger prediction sets in adaptive conformal prediction (CP).
- Then, the second point you make — about high-confidence predictions (resulting from low temperatures) — can be more clearly situated in this context.
- There are several places in the abstract where a prediction set is described as “efficient” or the word “efficiency” is otherwise employed to the same effect. I think this term is far too vague and could be conflated with computational efficiency to the casual reader.

### Introduction
- Referring to temperature scaling (Guo et al., 2017) and BNNs (Smith, 2013), it is written that “these approaches lack theoretical guarantees of model performance.”
  - It is unclear what exactly is meant by this.
- The next sentence sets up a vague contrast: “conformal prediction, on the other hand, …”
  - This contrast would benefit from being made more precise.

### Definition of Marginal Coverage (around Equation 2)
- It is not specified what the probability is over, making it impossible for readers to understand what is “marginal” about marginal coverage.

### Missing Definition of Conditional Coverage
- Immediately below Equation (4), it is written that the paper will focus on adaptive CP, which is “designed to improve conditional coverage.”
- If this is your focused notion of coverage, it is odd that there is no accompanying definition, especially when one is provided for marginal coverage in Equation (2).

### Consistent Usage of Marginal vs. Conditional Coverage
- By the end of Section 2, the reader understands there are two distinct notions of coverage: marginal and conditional.
- When you make statements henceforth such as “Notably, both methods incorporate a uniform random variable \( u \) to achieve exact \( 1 - \alpha \) coverage,” it seems important to clarify which type of coverage is being referred to.
- There are multiple other such instances. If the remainder of the paper will only consider conditional coverage, please state this explicitly and note that “coverage” will refer to that notion from then on.

### Section 3.1: Definition of Expected Calibration Error (ECE)
- The definition of the ECE at the beginning of Section 3.1 is incomplete for readers not already familiar with it.
- Either define it completely or omit the partial definition and instead reference a standard source.

### Table 1
- The triangles indicating superior/inferior performance are rather distracting, especially since the directionality of the arrows is reversed for ECE and Avg Size metrics.
- It would be helpful to reconsider the use of these symbols altogether especially you use the convention of boldface later on to express superiority.

- Additionally, I’m not familiar with all the confidence calibration methods listed.
  - Do they all involve choosing some type of temperature parameter?
  - If so, then the typical value of the temperature should be reported so that we can observe the tendency to select high-temperature (low-confidence) predictions.
  - This would help set up Section 3.2 more clearly and help the reader appreciate why you study low-temperature (high-confidence) predictions.

### Extensions to Other Post-Hoc Calibration Methods
- It is written that ConfTS can be extended to Platt scaling and vector scaling.
- What about the other calibration methods in Table 1?
- Can you set out more general conditions on the calibration method under which ConfTS can be applied?

### ConfTR
- I was surprised to learn about an important related work, ConfTR, only on page 10 in a short paragraph within the experiments section.
- It was even more confusing to learn that ConfTS can be trained with either the ConfTR loss or the ConfTS loss.
- If the method ConfTS can be defined via alternative losses, then this suggests a level of modularity that is not reflected in the naming. This modularity should be clarified earlier in the paper.

### Pseudo-Algorithms
- I think it will be quite helpful to include pseudo-algorithms for your ConfTS/PS/VS methods.
- These should illustrate how the loss function employed (e.g., ConfTR loss vs. ConfTS loss) affects the method.

**Strengths And Weaknesses:**

## Strengths and Weaknesses

The core idea of the paper is conveyed in a logical and coherent order:

- An initial empirical study reveals that the commonly held belief — that standard calibration methods should be used in combination with conformal prediction (CP) — might be misfounded if **interval size** is of concern. (And of course it should be: completely trivial sets can achieve nominal coverage.)
- It appears that existing calibration methods tend to produce lower-confidence predictions — although this is not explicitly stated (see my related requested change).
- This motivates the next section, where an empirical study on **high-confidence predictions** (via low-temperature settings in temperature scaling) shows that tighter prediction sets can be obtained.
- A theoretical explanation is then provided to account for this phenomenon — again, in the **specific context of temperature scaling**.
- The paper concludes by proposing a new loss function for selecting the calibration temperature, introducing what is effectively a **new family of methods**.
- Finally, an extensive empirical study is conducted on these methods. The empirical performance is very convincing.

That said, there is **insufficient emphasis** on the fact that both the empirical study in Section 3.2 and the theoretical study in Section 3.3 pertain **only to temperature scaling**.

This weakens the accuracy of broad claims such as the one made in the first bullet point of the contributions:

> “We discover that current confidence calibration methods typically lead to larger prediction sets in adaptive conformal prediction, while high-confidence predictions (using small temperatures) can enhance the efficiency of prediction sets.”

You provide evidence for the second clause — that high-confidence predictions yield tighter sets — in the context of **temperature scaling, Platt scaling, and vector scaling**. This has **not** been demonstrated for the **other calibration methods** explored in Table 1. These distinctions should be made clearer to avoid overgeneralization of your findings.

---

> ### Author Response · Authors · 2025-04-26
>
> We sincerely appreciate the reviewer's feedback and suggestions.
>
>
> > 1. The paper’s claim in Section 3.2 and 3.3 is based primarily on temperature scaling, lacking evidence for other calibration methods listed in Table 1.
>
> Thank you for your insightful comment. In Sections 3.2 and 3.3, we focus on temperature scaling due to its simplicity (involving only a single parameter $T$) and widespread use in confidence calibration. This choice allowed us to derive a rigorous theoretical proof to explain the phenomenon observed in Table 1. Specifically, our analysis demonstrates that high confidence caused by low temperature can shorten the prediction size in conformal prediction. This can be generalized to other calibration methods as they generally cause lower confidence in practice. Through the empirical and theoretical analysis in sections 3.2 and 3.3, we provide a clear insight into the relationship between conformal prediction and confidence calibration through the lens of temperature scaling.
>
> ### Abstract
>
> > 2. The paper should emphasize that current calibration methods result in higher temperatures.
>
> Thank you for the suggestion. We guess the reviewer suggested that "current calibration methods result in lower confidence", as the temperature value is only adopted in temperature scaling. We update the writing of the abstract accordingly in the revised version.
>
> > 3. The paper should emphasize high-confidence prediction results from low temperatures.
>
> Thank you for the suggestion. We update the abstract in the revised version accordingly.
>
> > 4. The word “efficient” or “efficiency” is vague.
>
> Thank you for the suggestion. In the revised version, we update the abstract and give clear definitions of these terms in Section 2 to ensure clarity.
>
> ### Introduction
>
> > 5. The statement “these approaches lack theoretical guarantees of model performance. Conformal prediction, on the other hand, …” is vague.
>
> Thank you for pointing out the ambiguous description. We update this sentence as "these approaches do not provide formal theoretical guarantees for the reliability of model predictions" [1].
>
> ### Definition of Marginal Coverage and Conditional Coverage
>
> > 6. The paper should specify what the probability of marginal coverage is over; the paper mentions conditional coverage without an accompanying definition; the paper should specify what "coverage" refers to.
>
> In the revised version, we address the writing concerns of coverage by adding a detailed description in Section 2 and a discussion of conditional coverage in Appendix M. In particular, the marginal coverage ensures that the prediction set includes the true label with at least $1-\alpha$ probability, on average across all test points. The probability is with respect to the randomness of the data sample $(X, Y)$. We use ‘coverage’ to represent ‘marginal coverage’ for convenience. In addition, we include a discussion on conditional coverage with a formal definition in Appendix K.
>
> ### Definition of Expected Calibration Error (ECE)
>
> > 7. The paper should define ECE completely.
>
> We add the definition of ECE in Section 3.1 of the revised version.
>
> ### Table 1:
>
> > 8. The triangles indicating superior/inferior performance are distracting.
>
> We'd like to clarify that the triangles are designed to avoid any misunderstanding of the large number of average set sizes. As readers may easily take a larger number as better performance, we add the triangles to emphasize the relative comparison with the baseline. We believe this element is critical for improving the clarity of the table.
>
> > 9. Do all the confidence calibration methods listed involve choosing some type of temperature parameter?
>
> Most post-hoc methods, such as Temperature Scaling (TS), Platt Scaling (PS), and Vector Scaling (VS), incorporate temperature parameters to adjust the confidence of final predictions. Specifically, TS and PS introduce one and two learnable parameters, respectively, while VS uses two vectors as learnable parameters. In contrast, training-based methods and Bayesian approaches typically do not rely on such parameters.
>
> ### Extensions to Other Post-Hoc Calibration Methods
>
> > 10. Can ConfTS extend to other calibration methods? What are the general conditions on the calibration method under which ConfTS can be applied?
>
> In the manuscript, we show that our method can be extended to post-hoc calibration methods, such as Platt scaling and Vector scaling. Generally, our method can be integrated with those post-hoc calibration methods with learnable parameters optimized by SGD. Practitioners can simply replace the NLL loss with our ConfTS loss, thereby automatically optimizing the parameters to enhance the efficiency of conformal prediction.
>
> ### References
>
> [1] Huang J, et al. Conformal prediction for deep classifier via label ranking. ICML, 2024.

---

> > ### Author Response · Authors · 2025-04-26
> >
> > ### ConfTR
> >
> > > 11. The paper should explain ConfTR in detail.
> >
> > Thank you for the suggestion. We add the detailed formulation of ConfTr in the revised version. In particular, ConfTr loss can be incorporated into our method as an alternative loss function. Our experimental results show that our loss function is superior to the ConfTr loss with smaller prediction sets.
> >
> > > 12. The paper should emphasize that ConfTS can work with different loss functions.
> >
> > Thank you for the suggestion. In the revised version, we add the description in Subsection 3.4: "the $\mathcal{L}_{\mathrm{ConfTS}}$ can be replaced by ConfTr loss or new training losses designed for different goals, e.g., improving conditional coverage."
> >
> > ### Pseudo-Algorithms
> >
> > > 13. The paper should include pseudo-algorithms to illustrate how the loss function employed affects the method.
> >
> > In the revised version, we add the pseudo-algorithms of ConfTS, ConfPS, and ConfVS in Appendix G.

---

### Review · Reviewer_UXE6 · 2025-04-15

**Summary Of Contributions:**

The paper presents a study of the interaction between conformal prediction set size efficiency and heuristic calibration methods in multiclass prediction settings. It takes the APS nonconformity score and temperature scaling as a starting point, and mathematically shows that as temperature decreases to 0, the average prediction set size under non-randomized APS goes down. Furthermore, it presents several experiments that further support this point, for both APS and RAPS nonconformity scores and for several other calibration methods (including Platt scaling and mixup).

Next, extending the insight of the relationship between the temperature and conformal set efficiency, it proposes a loss function and method (that consists in minimizing this loss) called ConfTS (conformal temperature scaling). This method (as well as its generalization that optimizes for the parameters of other calibration methods such as Platt scaling) is then empirically tested on a suite of multiclass experiments, showcasing the improvements in conformal set efficiency achieved by ConfTS.

**Audience:**

Yes

**Claims And Evidence:**

Yes

**Requested Changes:**

As one "weakness", I would like to point out the empirical evaluation of ConfTS. Specifically, I would like to see how the proposed method ConfTS affects various calibration metrics of the underlying models, in comparison to optimizing the temperature/the parameters of the calibration methods for the sake of calibration error decrease. In other words, what is the calibration suboptimality incurred by ConfTS in order to achieve improvements in the prediction set sizes? This could be fruitfully illustrated, for the benefit of the reader, by e.g. depicting some Pareto frontiers showing the tradeoff between ECE and prediction set sizes as a function of the parameter (e.g. temperature).

Furthermore, admittedly the experiments presented (especially at the beginning of the paper before ConfTS was introduced) were executed on somewhat old tasks. That could also be improved, especially as it is empirically known that calibration of NNs out of the box has gone up since Guo et al's paper in 2017.

**Strengths And Weaknesses:**

This manuscript explores an important area where two UQ paradigms, calibration and conformal prediction, fruitfully overlap. Specifically, it explores the effect of calibrating a multiclass predictor ,via one of the oft-used heuristic post-hoc calibration methods, on the size of downstream conformal prediction sets, when scores like APS and RAPS (which directly take in model probabilities) are sued. This question has to my knowledge somewhat flown under the radar, in comparison to the more direct question of optimizing the model outputs for the purposes of conformal prediction efficiency (as developed in Stutz et al (2022) and follow-up works). As such, I believe this paper is a timely contribution.

The method ConfTS is conceptually very straightforward, and yet is shown to provide significant efficiency increases in almost all of the performed experiments. Moreover, the authors show that it can even improve over the method of Stutz et al, which employs a generally more complex pipeline to obtain its results. This simplicity should lead to good illustrative value for the readers interested in calibration methods that can improve prediction set sizes.

On the theory side, the paper demonstrates the following simple fact: taking the sum of the softmax vector entries corresponding to top-k predicted probabilities (where k and the probabilities are fixed), driving the temperature down monotonically decreases this sum. Under some regularity assumptions, this then leads to the conclusion that the efficiency of the prediction sets under APS can be tuned by tuning the temperature. While the softmax fact is common knowledge in the intuitive sense that as t -> 0, the prediction set will converge to including only the top label due to the probability mass concentrating there, but to my best knowledge this fact has not been explicitly stated/derived before. Thus, even though this fact is quite simple (as a matter of fact, it can be obtained simply by taking the derivative of the softmax, so the longer argument in the paper is not necessary) but I do think it adds some value to the literature to have it written down in this form.

Furthermore, the paper is quite well-written, and does a good job explaining the setup and executing on the experiments.

To sum up, while the paper's contributions are quite simple and straightforward at first glance, to my knowledge such a study has not been explicitly performed before, and coupled with the paper's writing it should have value for the TMLR audience.

---

> ### Author Response · Authors · 2025-04-26
>
> We sincerely thank the reviewer for the feedback.
>
> > 1. The paper does not include an evaluation of how the proposed ConfTS method affects calibration metrics, such as the tradeoff between calibration error and prediction set sizes.
>
> Thank you for the suggestion. In the revised version, we present the ECE results of ConfTS, ConfPS and ConfVS in Appendix L. The results in Table 13 show that the parameters optimized by our methods achieve much smaller prediction sets than the baseline. It is also intuitive to see that the original version of TS, PS, and VS achieves lower ECE than the baseline and our methods, as our methods are not designed for confidence calibration. Notably, these two objectives do not conflict in practice as we can use different values of parameters (e.g., temperature) according to our goals in inference. To avoid any misunderstanding, we also include a clarification at the end of Section 3.
>
> > 2. The experiments are based on tasks considered somewhat outdated, which might not reflect the latest advancements in neural network calibration.
>
> We appreciate the reviewer’s feedback. To clarify, the tasks in Section 3 were selected to systematically analyze the interplay between conformal prediction and confidence calibration, rather than to benchmark the latest advancements in neural network calibration. Notably, we focus on **conformal prediction in classification tasks**. Thus, we use these established tasks (like CIFAR-100 and ImageNet) that remain widely used in conformal prediction [1,2].
>
> Moreover, our experiments employ both post-hoc and training-based calibration methods to ensure a comprehensive evaluation. Temperature scaling, in particular, is included as a representative post-hoc method due to its simplicity, effectiveness, and continued adoption in modern large-scale models (e.g., large language model [3] and vision-language model [4]). Thus, our experimental setup is both relevant and well-aligned with the goals of studying conformal prediction’s interaction with confidence calibration.
>
> ### References
> [1] Yan G, et al. Provably robust conformal prediction with improved efficiency. ICLR, 2024.
>
> [2] Huang J, et al. Conformal prediction for deep classifier via label ranking. ICML, 2024.
>
> [3] Xie J, et al. Calibrating language models with adaptive temperature scaling. EMNLP, 2024.
>
> [4] Wang S et al. Open-vocabulary calibration for fine-tuned CLIP. ICML, 2024.

---

### Decision · Action_Editor_jE2D · 2025-06-05

**Recommendation:** Accept with minor revision

**Additional Comments:**

One reviewer, in their final recommendation to me, had some issues about the claims made in this paper being too general, and I believe these issues are valid. First, let me me share these points in direct quotation, since the authors cannot see the official recommendations.

> After engaging with the authors during the review period, I find the claims in this paper to be generally supported but presented in a way that is suboptimal and at times misleading. Since they focus on the APS and RAPS score functions, they should be more explicit that the goal of these score function is to achieve approximate conditional coverage. With this in mind, in order to show that their proposed methods have value, they must show that they reduce set size while preserving conditional coverage. After making these suggestions to the author, they included results for conditional-coverage in the Appendix, which I appreciate. However, I believe the story would make more sense if these results were included in the main paper.

> When I say "misleading", I refer to sweeping statements such as "Confidence calibration methods deteriorate the efficiency of prediction sets" (pg. 4). It is very important to emphasize that such claims only apply to APS and RAPS, and do not apply to the most common classification score function, LAC. In my opinion, a honest description of the contribution of this paper is "a way to modify APS and RAPS to achieve smaller sets, without hurting the conditional coverage" -- I believe this would be of interest to TMLR readers. My main qualm is that the paper comes across as claiming more than this.

While the authors have mentioned how their insights do not hold under LAC in the paper, at present, this is more of a footnote, relegated to the "Limitations" paragraph at the end of the main text and the appendix. I would like to ask the authors to address the above point by making the limitations clear when describing the main claims in section 1.

In addition, since the paper is in essence about achieving better tradeoffs, I fully agree with the above reviewer on the point that conditional coverage should be discussed in the main paper; please work appendix M into the main paper so that the overall story is clear and honest.

The above two points are conditions for acceptance; please revise the paper accordingly.

**Audience:**

Yes

**Audience Explanation:**

The paper is well-written, the results are clear, and the methods being considered (both APS and confidence calibration) are widely applicable general techniques. The new method proposed in this paper shares this generality, and all the reviewers felt that this paper will have an audience in the TMLR community; I agree with them, and vote to accept the paper with minor revisions to be described below.

**Claims And Evidence:**

Yes

**Claims Explanation:**

Overall, the main claims center around how for some conformal prediction methods (namely, (R)APS), seeking better calibration through temperature-based confidence control can lead to larger prediction sets (i.e., "worse" performance in the conformal task, all else equal). The authors provide both empirical and theoretical evidence for this claim, and derive a new method for improving the "efficiency" of APS methods, namely achieving tighter sets without sacrificing the desired coverage guarantees. Regarding the validity of these claims and the overall clarity of the paper, the reviewers are positive on this paper, and I agree with them.

That said, one reviewer has requested that the limitations of the proposed method and current insights be made more explicit to the readers, another point I am in agreement with. Please see the "Additional comments" field below for requested changes.